# Offline Meta Reinforcement Learning – Identifiability Challenges and Effective Data Collection Strategies

**Ron Dorfman**
Technion
rdorfman@campus.technion.ac.il

**Idan Shenfeld**
Technion
idanshen@campus.technion.ac.il

**Aviv Tamar**
Technion
avivt@technion.ac.il

## Abstract

Consider the following instance of the Offline Meta Reinforcement Learning (OMRL) problem: given the complete training logs of $N$ conventional RL agents, trained on $N$ different tasks, design a meta-agent that can quickly maximize reward in a new, unseen task from the same task distribution. In particular, while each conventional RL agent explored and exploited its own different task, the meta-agent must identify regularities in the data that lead to effective exploration/exploitation in the unseen task. Here, we take a Bayesian RL (BRL) view, and seek to learn a Bayes-optimal policy from the offline data. Building on the recent VariBAD BRL approach, we develop an off-policy BRL method that learns to plan an exploration strategy based on an adaptive neural belief estimate. However, learning to infer such a belief from offline data brings a new identifiability issue we term MDP ambiguity. We characterize the problem, and suggest resolutions via data collection and modification procedures. Finally, we evaluate our framework on a diverse set of domains, including difficult sparse reward tasks, and demonstrate learning of effective exploration behavior that is qualitatively different from the exploration used by any RL agent in the data. Our code is available online at https://github.com/Rondorf/BOReL.

## 1 Introduction

A central question in reinforcement learning (RL) is how to learn quickly (i.e., with few samples) in a new environment. Meta-RL addresses this issue by training an agent on a large set of training environments [5, 9]. Intuitively, the meta-RL agent can learn regularities in the environments, which allow quick learning in any environment that shares a similar structure. Indeed, recent work demonstrated this by training memory-based controllers that 'identify' the domain [5, 28, 18], or by learning a parameter initialization that leads to good performance within a few gradient steps [9].

Another formulation of quick RL is Bayesian RL [BRL, 11]. In BRL, the environment parameters are treated as unobserved variables, with a known prior distribution. Consequentially, the standard problem of maximizing expected returns (taken with respect to the posterior distribution) *explicitly accounts for the environment uncertainty*, and its solution is a *Bayes-optimal* policy, wherein actions optimally balance exploration and exploitation. Recently, Zintgraf et al. [36] showed that meta-RL is in fact an instance of BRL, where the meta-RL environment distribution is simply the BRL prior. Furthermore, a Bayes-optimal policy can be trained using standard RL methods, by adding to the state the belief over the environment parameters. The VariBAD algorithm [36] implements this idea using a variational autoencoder (VAE) for belief estimation and deep neural network policies.

35th Conference on Neural Information Processing Systems (NeurIPS 2021).

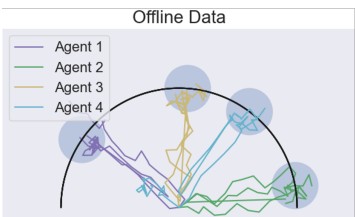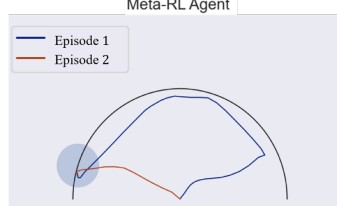

Figure 1: Offline meta-RL on the Semi-Circle domain: the task is to navigate to a goal position that can be anywhere on the semi-circle. The reward is sparse (light-blue), and the offline data (left) contains training logs of conventional RL agents trained to find individual goals. The meta-RL agent (right) needs to find a policy that quickly finds the unknown goal, here, by searching across the semi-circle in the first episode, and directly reaching it the second – a completely different strategy from the dominant behaviors in the data.

Most meta-RL studies, including VariBAD, have focused on the *online* setting, where, during training, the meta-RL policy is continually updated using data collected from running it in the training environments. In domains where data collection is expensive, such as robotics and healthcare to name a few, online training is a limiting factor. For standard RL, offline (a.k.a. batch) RL mitigates this problem by learning from data collected beforehand by an arbitrary policy [7, 21]. In this work we investigate the *offline approach to meta-RL* (OMRL).

Any offline RL approach is heavily influenced by the data collection policy. To ground our investigation, we focus on the following practical setting:[1] we assume that data is collected by running standard RL agents on a set of environments from the environment distribution. While the data was not specifically collected for the meta-RL task, we hypothesize that regularities between the training domains can still be learned, to provide faster learning in new environments. Figure 1 illustrates our problem: in this navigation task, each RL agent in the data learned to find its own goal, and converged to a behavior that quickly navigates toward it. The meta-RL agent, on the other hand, needs to learn a completely different behavior that effectively *searches* for the unknown goal position.

Our method for solving OMRL is an off-policy variant of the VariBAD algorithm, based on replacing the on-policy policy gradient optimization in VariBAD with an off-policy Q-learning based approach. This, however, requires some care, as Q-learning applies to states of fully observed systems. We show that the VariBAD approach of augmenting states with the belief in the data applies to the off-policy setting as well, leading to an effective and practical algorithm. The offline setting, however, brings about another challenge – when the agent visits different parts of the state space in different environments, learning to identify the correct environment and obtain an accurate belief estimate becomes challenging, a problem we term *MDP ambiguity*. We formalize this problem, and discuss how it manifests in common scenarios such as sparse rewards or sparse differences in transitions. Based on our formalization, we propose a general data collection strategy that can mitigate the problem. Furthermore, when ambiguity is only due to reward differences, we show that a simple reward relabelling trick suffices, without changing data collection. We collectively term our data collection/relabelling and off-policy algorithm as *Bayesian Offline Reinforcement Learning (BOReL)*.

In our experiments, we show that BOReL learns effective exploration policies from offline data on both discrete and continuous control problems. We demonstrate significantly better exploration than meta-RL methods based on Thompson sampling such as PEARL [28], *even when these methods are allowed to train online*. Furthermore, we explore the issue of MDP ambiguity in practice, and demonstrate that, when applicable, our proposed solutions successfully mitigate it.

An important implication of our study is that without suitably collected data, MDP ambiguity can make learning an effective offline BRL solution impossible. This stands in contrast to recent studies in offline (non-Bayesian) RL [30, 10, 19], where the focus is on designing learning algorithms for arbitrarily collected data, typically by avoiding actions that were not explored enough. Outside offline RL, however, it is common to develop data collection methods based on formal limitations of the problem. For example, the use of randomized controlled trials in medical treatments is well motivated by the theory of causal inference [27]. The data collection methods we propose here are simple and practical to implement, and as we demonstrate, effectively handle MDP ambiguity.

---

[1]The theory and algorithms we develop, however, *are not limited* to any particular data collection protocol.

Our main contributions are therefore as follows: to our knowledge, this is the first study of meta learning exploration in the offline setting; we provide the necessary theory to extend VariBAD to off-policy RL; we formulate MDP ambiguity, which characterizes which problems are solvable under the offline BRL setting, and based on this formulation we propose several principled data collection strategies; we show non-trivial empirical results that demonstrate significantly better exploration than meta-RL methods based on Thompson sampling; finally, and of independent interest, our off-policy algorithm significantly improves the sample efficiency of conventional VariBAD in the online setting.

## 2   Background

We recapitulate meta-RL, BRL and the VariBAD algorithm.

**Meta-RL:** In meta-RL, a distribution over tasks is assumed. A task $\mathcal{T}_i$ is described by a Markov Decision Process [MDP, 2] $\mathcal{M}_i = (\mathcal{S}, \mathcal{A}, \mathcal{R}_i, \mathcal{P}_i)$, where the state space $\mathcal{S}$ and the action space $\mathcal{A}$ are shared across tasks, and $\mathcal{R}_i$ and $\mathcal{P}_i$ are task specific reward and transition functions. Thus, we write the task distribution as $p(\mathcal{R}, \mathcal{P})$. For simplicity, we assume throughout that the initial state distribution $P_{init}(s_0)$ is the same for all MDPs. The goal in meta-RL is to train an agent that can quickly maximize reward in new, unseen tasks, drawn from $p(\mathcal{R}, \mathcal{P})$.

**Bayesian Reinforcement Learning:** The goal in BRL is to find the optimal policy $\pi$ in an MDP, when the transitions and rewards are not known in advance. Similar to meta-RL, we assume a prior over the MDP parameters $p(\mathcal{R}, \mathcal{P})$, and seek to maximize the expected discounted return,

$$\mathbb{E}_\pi \left[ \sum_{t=0}^\infty \gamma^t r(s_t, a_t) \right], \tag{1}$$

where the expectation is taken with respect to *both the uncertainty in state-action transitions $s_{t+1} \sim \mathcal{P}(\cdot | s_t, a_t)$, $a_t \sim \pi$, and the uncertainty in the MDP parameters $\mathcal{R}, \mathcal{P} \sim p(\mathcal{R}, \mathcal{P})$.*[2] Key here is that this formulation naturally accounts for the exploration/exploitation tradeoff – an optimal agent must plan its actions to reduce uncertainty in the MDP parameters, if such leads to higher rewards.

One way to approach the BRL problem is to model $\mathcal{R}, \mathcal{P}$ as unobserved state variables in a partially observed MDP [POMDP, 3], reducing the problem to solving a particular POMDP instance where the unobserved variables do not change in time. The *belief* at time $t$, $b_t$, denotes the posterior probability over $\mathcal{R}, \mathcal{P}$ given the history of state transitions and rewards observed until this time $b_t = P(\mathcal{R}, \mathcal{P} | h_{:t})$, where $h_{:t} = \{s_0, a_0, r_1, s_1 \ldots, r_t, s_t\}$ (note that we denote the reward after observing the state and action at time $t$ as $r_{t+1} = r(s_t, a_t)$). The belief can be updated iteratively according to Bayes rule, where $b_0(\mathcal{R}, \mathcal{P}) = p(\mathcal{R}, \mathcal{P})$, and: $b_{t+1}(\mathcal{R}, \mathcal{P}) = P(\mathcal{R}, \mathcal{P} | h_{:t+1}) \propto P(s_{t+1}, r_{t+1} | h_{:t}, \mathcal{R}, \mathcal{P}) b_t(\mathcal{R}, \mathcal{P})$.

Similar to the idea of solving a POMDP by representing it as an MDP over belief states, the state in BRL can be augmented with the belief to result in the Bayes-Adaptive MDP model [BAMDP, 6]. Denote the augmented state $s_t^+ = (s_t, b_t)$ and the augmented state space $\mathcal{S}^+ = \mathcal{S} \times \mathcal{B}$, where $\mathcal{B}$ denotes the belief space. The transitions in the BAMDP are given by: $P^+(s_{t+1}^+ | s_t^+, a_t) = \mathbb{E}_{b_t} [\mathcal{P}(s_{t+1} | s_t, a_t)] \delta(b_{t+1} = P(\mathcal{R}, \mathcal{P} | h_{:t+1}))$, and the reward in the BAMDP is the expected reward with respect to the belief: $R^+(s_t^+, a_t) = \mathbb{E}_{b_t} [\mathcal{R}(s_t, a_t)]$. The Bayes-optimal agent seeks to maximize the expected discounted return in the BAMDP, and the optimal solution of the BAMDP gives the optimal BRL policy. As in standard MDPs, the optimal action-value function in the BAMDP satisfies the Bellman equation: $\forall s^+ \in \mathcal{S}^+, a \in \mathcal{A}$ we have that

$$Q(s^+, a) = R^+(s^+, a) + \gamma \mathbb{E}_{s^{+'} \sim P^+} \left[ \max_{a'} Q(s^{+'}, a') \right]. \tag{2}$$

Computing a Bayes-optimal agent amounts to solving the BAMDP, where the optimal policy is a function of the augmented state. For most problems this is intractable, as the augmented state space is continuous and high-dimensional, and the posterior update is also intractable in general.

**The VariBAD Algorithm:** VariBAD [36] approximates the Bayes-optimal solution by combining a model for the MDP parameter uncertainty, and an optimization method for the corresponding BAMDP. The MDP parameters are represented by a vector $m \in \mathbb{R}^d$, corresponding to the latent variables in a

---

[2]For ease of presentation, we consider the infinite horizon discounted return. Our formulation easily extends to the episodic and finite horizon settings, as considered in our experiments.

parametric generative model for the state-reward trajectory distribution conditioned on the actions $P(s_0, r_1, s_1 \ldots, r_H, s_H | a_0, \ldots, a_{H-1}) = \int p_\theta(m) p_\theta(s_0, r_1, s_1 \ldots, r_H, s_H | m, a_0, \ldots, a_{H-1}) dm$. The model parameters $\theta$ are learned by a variational approximation to the maximum likelihood objective, where the variational approximation to the posterior $P(m | s_0, r_1, s_1 \ldots, r_H, s_H, a_0, \ldots, a_{H-1})$ is chosen to have the structure $q_\phi(m | s_0, a_0, r_1, s_1 \ldots, r_t, s_t) = q_\phi(m | h_{:t})$. That is, the approximate posterior is conditioned on the history up to time $t$. The evidence lower bound (ELBO) in this case is $ELBO_t = \mathbb{E}_{m \sim q_\phi(\cdot | h_{:t})} [\log p_\theta(s_0, r_1, s_1 \ldots, r_H, s_H | m, a_0, \ldots, a_{H-1})] - D_{KL}(q_\phi(m | h_{:t}) || p_\theta(m))$. The main claim of Zintgraf et al. [36] is that $q_\phi(m | h_{:t})$ can be taken as an approximation of the belief $b_t$. In practice, $q_\phi(m | h_{:t})$ is represented as a Gaussian distribution $q(m | h_{:t}) = \mathcal{N}(\mu(h_{:t}), \Sigma(h_{:t}))$, where $\mu$ and $\Sigma$ are learned recurrent neural networks. While other neural belief representations could be used [15], we chose to focus on VariBAD for concreteness.

To approximately solve the BAMDP, [36] exploit the fact that an optimal BAMDP policy is a function of the state and belief, and therefore consider neural network policies that take the augmented BAMDP state as input $\pi(a_t | s_t, q_\phi(m | h_{:t}))$, where the posterior is practically represented by the distribution parameters $\mu(h_{:t}), \Sigma(h_{:t})$. The policies are trained using policy gradients, optimizing

$$J(\pi) = \mathbb{E}_{\mathcal{R}, \mathcal{P}} \mathbb{E}_\pi \left[ \sum_{t=0}^{H} \gamma^t r(s_t, a_t) \right]. \tag{3}$$

The expectation over MDP parameters in (3) is approximated by averaging over training environments, and the RL agent is trained online, alongside the VAE.

## 3 OMRL and Off-Policy VariBAD

In this section, we derive an off-policy variant of the VariBAD algorithm, and apply it to the OMRL problem. We first describe OMRL, and then present our algorithm.

### 3.1 OMRL Problem Definition

We follow the Meta-RL and BRL formulation described above, with a prior distribution over MDP parameters $p(\mathcal{R}, \mathcal{P})$. We are provided training data of an agent interacting with $N$ different MDPs, $\{\mathcal{R}_i, \mathcal{P}_i\}_{i=1}^N$, sampled from the prior. Each interaction is organized as $M$ trajectories of length $H$, $\tau^{i,j} = s_0^{i,j}, a_0^{i,j}, r_1^{i,j}, s_1^{i,j} \ldots, r_H^{i,j}, s_H^{i,j}$, $i \in 1, \ldots, N, j \in 1, \ldots, M$, where the rewards satisfy $r_{t+1}^{i,j} = \mathcal{R}_i(s_t^{i,j}, a_t^{i,j})$, the transitions satisfy $s_{t+1}^{i,j} \sim \mathcal{P}_i(\cdot | s_t^{i,j}, a_t^{i,j})$, and the actions are chosen from an arbitrary data collection policy. To ground our work in a specific context, we sometimes assume that the trajectories are obtained from running a conventional RL agent in each one of the MDPs (i.e., the complete RL training logs), which implicitly specifies the data collection policy. We will later investigate implications of this assumption, but emphasize that *this is merely an illustration, and our approach does not place any such constraint* – the trajectories can also be collected differently. Our goal is to use the data for learning a Bayes-optimal policy, i.e., a policy $\pi$ that maximizes Eq. (1).

### 3.2 Off-Policy VariBAD

The on-policy VariBAD algorithm cannot be applied to our offline setting. Our first step is to modify VariBAD to work off-policy. We start with an observation about the use of the BAMDP formulation in VariBAD, which will motivate our subsequent development.

**Does VariBAD really optimize the BAMDP?** Recall that a BAMDP is in fact a reduction of a POMDP to an MDP over augmented states $s^+ = (s, b)$, and with the rewards and transitions given by $R^+$ and $P^+$. Thus, an optimal Markov policy for the BAMDP exists in the form of $\pi(s^+)$. The VariBAD policy, as described above, similarly takes as input the augmented state, and is thus capable of representing an optimal BAMDP policy. However, *VariBAD's policy optimization in Eq. (3) does not make use of the BAMDP parameters $R^+$ and $P^+$!* While at first this may seem counterintuitive, Eq. (3) is in fact a sound objective for the BAMDP, as we now show.[3]

**Proposition 1.** *Let $\tau = s_0, a_0, r_1, s_1 \ldots, r_H, s_H$ denote a random trajectory from a fixed history dependent policy $\pi$, generated according to the following process. First, MDP parameters $\mathcal{R}, \mathcal{P}$*

---

[3]This result is closely related to the discussion in [14, 26], here applied to our particular setting.

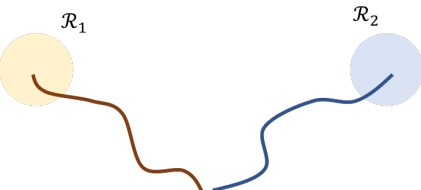

Figure 2: Reward ambiguity: from the two trajectories, it is impossible to know if there are two MDPs with different rewards (blue and yellow circles), or one MDP with rewards at both locations.

*are drawn from the prior $p(\mathcal{R}, \mathcal{P})$. Then, the state trajectory is generated according to $s_0 \sim P_{init}$, $a_t \sim \pi(\cdot|s_0, a_0, r_1, \ldots, s_t)$, $s_{t+1} \sim \mathcal{P}(\cdot|s_t, a_t)$ and $r_{t+1} \sim \mathcal{R}(s_t, a_t)$. Let $b_t$ denote the posterior belief at time t, $b_t = P(\mathcal{R}, \mathcal{P}|s_0, a_0, r_1, \ldots, s_t)$. Then, $P(s_{t+1}|s_0, a_0, r_1, \ldots, r_t, s_t, a_t) = \mathbb{E}_{\mathcal{R},\mathcal{P}\sim b_t}P(s_{t+1}|s_t, a_t)$, and $P(r_{t+1}|s_0, a_0, r_1, \ldots, s_t, a_t) = \mathbb{E}_{\mathcal{R},\mathcal{P}\sim b_t}\mathcal{R}(r_{t+1}|s_t, a_t)$.*

For on-policy VariBAD, Proposition 1 shows that the rewards and transitions in each trajectory can be seen as sampled from a distribution that **in expectation** is equal to $R^+$ and $P^+$, and therefore maximizing Eq. 3 is valid.[4] However, off-policy RL does not take as input trajectories, but tuples of the form $(s, a, r, s') \equiv (state, action, reward, next\ state)$, where states and actions can be sampled **from any distribution**. For an arbitrary distribution of augmented states, we must replace the rewards and transitions in our data with $R^+$ and $P^+$, which can be difficult to compute. Fortunately, Proposition 1 shows that when collecting data by sampling complete trajectories and corresponding beliefs, this is not necessary, as in expectation, the rewards and transitions are correctly sampled from the BAMDP. In the following, we therefore focus on settings where data can be collected that way, for example, by collecting logs of RL agents trained on the different training tasks.

Based on Proposition 1, we can use a state augmentation method similar to VariBAD, which we refer to as **state relabelling**. Consider each trajectory in our data $\tau^{i,j} = s_0^{i,j}, a_0^{i,j}, r_1^{i,j}, \ldots, s_H^{i,j}$, as defined above. Recall that the VariBAD VAE encoder provides an estimate of the belief given the state history $q(m|h_{:t}) = \mathcal{N}(\mu(h_{:t}), \Sigma(h_{:t}))$. Thus, we can run the encoder on every partial $t$-length history $\tau_{:t}^{i,j}$ to obtain the belief at each time step. Following the BAMDP formulation, we define the augmented state $s_t^{+,i,j} = (s_t^{i,j}, b_t^{i,j})$, where $b_t^{i,j} = \mu(\tau_{:t}^{i,j}), \Sigma(\tau_{:t}^{i,j})$. We next replace each state in our data $s_t^{i,j}$ with $s_t^{+,i,j}$, effectively transforming the data to as coming from a BAMDP. After applying state relabelling, any off-policy RL algorithm can be applied to the modified data, for learning a Bayes-optimal policy. In our experiments we used deep Q-learning [DQN, 25] for discrete action domains, and soft actor critic [SAC, 17] for continuous control.

## 4 Identifiability Problems in OMRL

We take a closer look at the OMRL problem. While in principle, it is possible to simply run off-policy VariBAD on the offline data, we claim that in many problems this may not work well. The reason is that the VariBAD belief update should reason about the uncertainty in the MDP parameters, which requires to effectively distinguish between the different possible MDPs. Training the VAE to distinguish between MDPs, however, *depends on the offline data*, and might not always be possible. This problem, which we term *MDP ambiguity*, is illustrated in Figure 2: consider two MDPs, one with rewards in the blue circle, and the other with rewards in the yellow circle. If the data contains trajectories similar to the ones in the figure, it is impossible to distinguish between having two different MDPs with the indicated rewards, or a single MDP with rewards at both the blue and yellow circles. Accordingly, we cannot expect to learn a meaningful belief update. In the following, we formalize MDP ambiguity, and how it can be avoided.

For an MDP defined by $\{\mathcal{R}, \mathcal{P}\}$, denote by $P_{\mathcal{R},\mathcal{P},\pi}(s, a, r, s')$ and $P_{\mathcal{P},\pi}(s, a)$ the distribution over $(s, a, r, s')$ and $(s, a)$, respectively, induced by a policy $\pi$.

**Definition 1** (MDP Ambiguity). *Consider data coming from a set of $N$ different MDPs $M = \{\mathcal{R}_i, \mathcal{P}_i\}_{i=1}^N \subset \mathcal{M}$, where $\mathcal{M}$ is an hypothesis set of possible MDPs, and corresponding data collection policies $\{\pi_\beta^i\}_{i=1}^N$, resulting in $N$ different data distributions $D = \{P_{\mathcal{R}_i,\mathcal{P}_i,\pi_\beta^i}(s, a, r, s')\}_{i=1}^N$.*

---

[4]To further clarify, if we could calculate $R^+$, replacing all rewards in the trajectories with $R^+$ will result in a lower variance policy update, similar to expected SARSA [34].

*We say that the data is ambiguous if there is an MDP $\{\mathcal{R}, \mathcal{P}\} \in \mathcal{M}$ and two policies $\pi$ and $\pi'$ such that $P_{\mathcal{R}_i, \mathcal{P}_i, \pi_\beta^i}(s, a, r, s') = P_{\mathcal{R}, \mathcal{P}, \pi}(s, a, r, s')$ and $P_{\mathcal{R}_j, \mathcal{P}_j, \pi_\beta^j}(s, a, r, s') = P_{\mathcal{R}, \mathcal{P}, \pi'}(s, a, r, s')$, for some $i \neq j$. Otherwise, the data is termed identifiable.*[5]

The essence of identifiability, as expressed in Definition 1, is that there is no single MDP in the hypothesis set that can explain data from multiple MDPs in the data, as in this case it will be impossible to learn an inference model that accurately distinguishes between the different MDPs, even with infinite data.[6] Let us now define the notion of *identifying state-actions* and *overlapping state-actions*, which will enable us to clearly state a simple sufficient condition for identifiability.

**Definition 2** (Identifying State-Action). *For a pair of MDPs $i$ and $j$, we say that $(s, a)$ is an identifying state-action pair if $\mathcal{R}_i(s, a) \neq \mathcal{R}_j(s, a)$ and/or $\mathcal{P}_i(s'|s, a) \neq \mathcal{P}_j(s'|s, a)$.*

**Definition 3** (Overlapping State-Action). *Consider the setting in Definition 1. For a pair of MDPs $i$ and $j$, we say that a state-action pair $(s, a)$ overlaps if it has positive probability under both $i$ and $j$, i.e., $P_{\mathcal{P}_i, \pi_\beta^i}(s, a), P_{\mathcal{P}_j, \pi_\beta^j}(s, a) > 0$.*

Identifiability strongly depends on the hypothesis set $\mathcal{M}$. However, for learning deep neural network inference models, we do not want to impose any structure on $\mathcal{M}$. Thus, in the following we provide a sufficient identifiability condition that holds for any $\mathcal{M}$.

**Proposition 2.** *Consider the setting in Definition 1. If for every $i \neq j$ there exists an identifying state-action pair that overlaps, then the data is identifiable.*

Thus, if the agent has data on identifying state-actions *obtained from different MDPs*, it has the capability to identify which data samples belong to which MDP. We next categorize several common types of meta-RL problems according to identifiability, as per Proposition 2; we will later revisit this dichotomy in our experiments. For our illustration, we assume that in each training MDP, the data collecting policy is approximately optimal (this would be the case when training standard RL agents on each MDP). Let us first consider problems that only differ in the reward. Here, when identifying state-actions (i.e., state-actions with different rewards) in different MDPs do not overlap, we will have an identifiability problem. The sparse reward tasks in Figures 1 and 2 are examples of this case – each agent will visit only its own reward area, resulting in ambiguity. When the rewards are dense, however, it is much more likely that the data is identifiable; common tasks like Half-Cheetah-Vel (cf. Sec. 6) are examples of this setting.[7] For MDPs that differ in their transitions, a similar argument can be made about whether the identifying state-actions overlap or not. Most studies on online and offline meta-RL to date considered problems with overlapping identifying state-actions, where ambiguity is not an issue. For example, in the Walker environment of Zintgraf et al. [36], the shape of the agent is varied, which manifests in almost every transition, and a successful agent must walk forward, thus many overlapping state-actions are visited; the Wind domain (cf. Sec. 6) is another example. Examples of problems with non-overlapping identifying transitions are, for example, peg-in-hole insertion where the hole position varies between tasks, or the Escape-Room domain in Sec. 6; in such domains we expect ambiguity to be a concern. One can of course imagine combinations and variations of the categories above – our aim is not to be exhaustive, but to illustrate which OMRL problems are difficult due to ambiguity, and which are not.

Note that MDP ambiguity is special to offline meta-RL; in online meta-RL, the agent may be driven by the online adapting policy (or guided explicitly) to explore states that reduce its ambiguity. We also emphasize that this problem is not encountered in standard (non-meta) offline RL, as the problem here concerns the *identification of the MDP*, which in standard RL is unique.

How can one collect data to mitigate MDP ambiguity? We present a simple, general modification to the data collection scheme we term **policy replaying**, which, under mild conditions on the original data collection policies, guarantees that the resulting data will be identifiable. We importantly note that changing the data collection method in-hindsight is not suitable for the offline setting. Therefore, the proposed scheme should be viewed as *a guideline for effective OMRL data collection*. For each

---

[5]$P(\cdot) = P'(\cdot)$ means equality almost everywhere; $P(\cdot) \neq P'(\cdot)$ means that equality almost everywhere does not hold.

[6]For simplicity, Definition 1 considers a discrete set of MDPs, and infinite data. In our experiments, we validate that our insights also hold for finite data and continuous models.

[7]The ambiguity of sparse reward tasks is very different from the well-known exploration difficulty in RL with sparse rewards (e.g., [1]): *ambiguity is not related to the RL algorithm but to the learned belief estimate*.

MDP, we propose collecting data in the following manner: randomly draw a data collection policy from $\{\pi_\beta^i\}_{i=1}^N$, collect a trajectory following that policy, and repeat. After this procedure, the new data distributions are all associated with *the same* data collection policy, which we denote $\pi_r$.

**Proposition 3.** *For every $i \neq j$, denote the set of identifying state-action pairs by $\mathcal{I}_{i,j}$. If for every $i$ and every $j$ exists $(s_{i,j}, a_{i,j}) \in \mathcal{I}_{i,j}$ such that $P_{\mathcal{P}_i, \pi_\beta^i}(s_{i,j}, a_{i,j}) > 0$, then replacing $\pi_\beta^i$ with $\pi_r$ for all $i$ results in identifiable data.*

Note that the requirement on identifying states in Proposition 3 is minimal – without it, the original data collecting policies $\pi_\beta^i$ are useless, as they do not visit any identifying states (e.g., consider the example in Figure 2 with policies that do not visit the reward at all).

When the tasks only differ in their reward function, and the reward functions for the training environments are known, policy replaying can be implemented in hindsight, *without changing the data collection process*. This technique, which we term **Reward Relabelling (RR)**, is applicable under the offline setting, and described next. In RR, we replace the rewards in a trajectory from some MDP $i$ in the data with rewards from another randomly chosen MDP $j \neq i$. That is, for each $i \in 1, \dots, N$, we add to the data $K$ trajectories $\hat{\tau}^{i,k}$, $k \in 1, \dots, K$, where $\hat{\tau}^{i,k} = (s_0^{i,k}, a_0^{i,k}, \hat{r}_1^{i,k}, s_1^{i,k} \dots, \hat{r}_H^{i,k}, s_H^{i,k})$, where the relabelled rewards $\hat{r}$ satisfy $\hat{r}_{t+1}^{i,k} = \mathcal{R}_j(s_t^{i,k}, a_t^{i,k})$. Thus, our relabelling effectively runs $\pi_\beta^i$ on MDP $j$, which is equivalent to performing policy replaying (in hindsight). We remark that the assumption on known reward (during training) is mild, as the reward is the practitioner's method of specifying the task goal, which is typically known [13, 32, 31]; this assumption is also satisfied in all meta-RL studies to date.

**BOReL**: we refer to the BOReL algorithm as the combination of the policy replaying/RR techniques and off-policy RL applied to state-relabelled trajectories. In Appendix B we provide pseudo-code, and detail how to apply the insights of Proposition 3 to a practical episodic RL setting.

# 5 Related Work

We focus on meta-RL – quickly learning to solve RL problems. Gradient-based approaches to meta-RL seek policy parameters that can be updated to the current task with a few gradient steps [9, 12, 29, 4]. These are essentially online methods, and several studies investigated learning of structured exploration strategies in this setting [16, 29, 33]. Memory-based meta-RL, on the other hand, map the observed history in a task $h_{:t}$ to an action [5, 35]. These methods effectively treat the problem as a POMDP, and learn a memory based controller for it.

The connection between meta-learning and Bayesian methods, and between meta-RL and Bayesian RL in particular, has been investigated in a series of recent papers [20, 18, 26, 36], and our work closely follows these ideas. In particular, these works elucidate the difference between Thompson-sampling based strategies, such as PEARL [28], and Bayes-optimal policies, such as VariBAD, and suggest to estimate the BAMDP belief using the latent state of deep generative models. *Our contribution is an extension of these ideas to the offline RL setting*, which to the best of our knowledge is novel. Technically, the VariBAD algorithm in [36] is limited to on-policy RL, and the off-policy method in [18] requires specific task descriptors during learning, while VariBAD, which our work is based on, does not. One can also learn neural belief models using contrastive learning [15]; our methods and identifiability discussion apply to this case as well.

**Concurrently and independently with our work**, Li et al. [22] proposed MBML, an offline meta-RL algorithm that combines BCQ [10] with a task inference module. Interestingly, Li et al. [22] describe a technique similar to reward relabelling for discouraging task inference to ignore rewards. Here, we provide a formal and general characterization of identifiability problems in OMRL. Additionally, MBML does not take into account task uncertainty, and cannot plan actions that actively explore to reduce this uncertainty – this is a form of Thompson sampling, where a task-conditional policy reacts to the task inference (see Figure 1 in [22]). **Our work is the first to tackle offline meta-learning of Bayes-optimal exploration**. In addition, we demonstrate the first offline results on sparse reward tasks, which, compared to the dense reward tasks in [22], require a significantly more complicated solution than Thompson sampling (see experiments section). We achieve this by building on BRL theory, which both optimizes for Bayes-optimality and results in a much simpler algorithm. Recent work on meta Q-learning [8] also does not incorporate task uncertainty, and thus cannot be Bayes-optimal. The very recent work of Mitchell et al. [24] considers a different offline meta-RL setting, where an offline dataset from the test environment is available.

Classical works on BRL are comprehensively surveyed by Ghavamzadeh et al. [11]. Our work, in comparison, allows training scalable deep BRL policies. Finally, there is growing interest in offline deep RL [30, 21]. In our experiments, a state-of-the-art method of this flavor (CQL, [19]) led to minor improvements, though future offline RL developments may possibly benefit OMRL too.

## 6 Experiments

In our experiments, we aim to demonstrate: (1) Learning approximately Bayes-optimal policies in the offline setting; and (2) The severity of MDP ambiguity, and the effectiveness of our proposed resolutions. In the supplementary material, we also report that our off-policy method improves meta-RL performance in the online setting.

Answering (1) is difficult because the Bayes-optimal policy is generally intractable, and because our results crucially depend on the available data. However, in deterministic domains with a single sparse reward, the optimal solution amounts to '*search all possible goal locations as efficiently as possible, and stay at goal once found; in subsequent episodes, move directly to goal*'. We therefore chose domains where this behavior can be identified qualitatively. Quantitatively, we compare BOReL with two Thompson sampling based methods: MBML [22] *using the same offline data,* and PEARL [28], *using online data*, and aim to show that the performance improvement due to being approximately Bayes-optimal gives an advantage, *even under the offline data* restriction.

**Domains and evaluation metric:** we evaluate learning to explore efficiently in a diverse set of domains: (1) A discrete $5 \times 5$ **Gridworld** [36]; (2) **Semi-circle** – a continuous point robot where a sparse reward is located somewhere on a semi-circle (see Figure 1); (3) **Ant-Semi-circle** – a challenging modification of the popular Ant-Goal task [8] to a sparse reward setting similar to the semi-circle task above (see Figure 5); (4) **Half-Cheetah-Vel** [9], a popular high-dimensional control domain with dense rewards; (5) **Reacher-Image** – 2-link robot reaching an unseen target located somewhere on a quarter circle, with image observations and dense rewards (see Appendix D); (6) **Wind** – a point robot navigating to a fixed goal in the presence of varying wind; and (7) **Escape-Room** – a point robot that needs to escape a circular room where the only opening is somewhere on the semi-circle (full details in Appendix D). These domains portray both discrete (1) and continuous (2-7) dynamics, and environments that differ either in the rewards (1-5) or transitions (6-7). Domains (3), (4) and (5) are high-dimensional, and the navigation problems (1-3, 7) require non-trivial exploration behavior to quickly identify the task. Importantly, relating to the MDP ambiguity discussion in Sec. 4, optimal policies for domains (1-3, 7) have non-overlapping identifying states; here we expect MDP ambiguity to be a problem. On the other hand, in domains (4-6) the identifying states are expected to overlap, as the rewards/transition differences are dense. To evaluate performance, we measure average reward in the first 2 episodes on unseen tasks – this is where efficient exploration makes a critical difference.[8] In the supplementary, we report results for more evaluation episodes.

**Data collection and organization:** For data collection, we used off-the-shelf DQN (Gridworld) and SAC (continuous domains) implementations. To study the effect of data diversity, we diversified the offline dataset by modifying the initial state distribution $P_{init}$

[8]For Gridworld, we measure average reward in the first 4 episodes, and for Wind, only in the first episode.

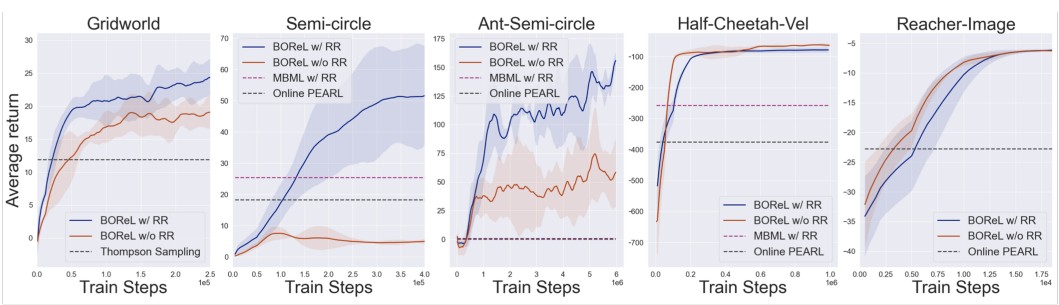

Figure 3: Offline performance on domains with varying rewards. We compare BOReL with and without reward relabeling (blue and red, respectively) with Thompson sampling baselines – calculated exactly in Gridworld, and using online PEARL and offline MBML for the other domains. Full training curves for baselines appear in the supplementary; here we plot only the best performance.

to either (1) uniform over a large region, (2) uniform over a restricted region, or (3) fixed to a single position. At meta-test time, only the single fixed position is used. The tasks are episodic, but we want the agent to maintain its belief between episodes, so that it can continually improve performance (see Figure 1). We follow Zintgraf et al. [36], and aggregate $k$ consecutive episodes of length $H$ to a long trajectory of length $k \times H$, and we do not reset the hidden state in the VAE recurrent neural network after episode termination. For reward relabelling, we replace either the first or last $k/2$ trajectories with trajectories from a randomly chosen MDP, and relabel their rewards. For policy replay we replace trajectories by sampling a new trajectory using the trained RL policy of another MDP. Technically, network architectures and hyperparameters were chosen similarly to [36], as detailed in the supplementary.

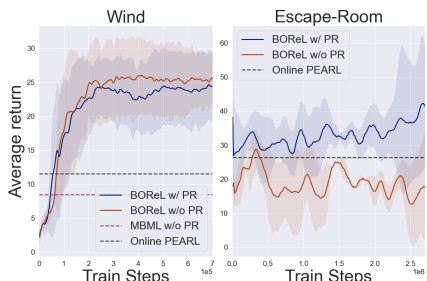

Figure 4: Offline performance on domains with varying transitions. We compare BOReL with and without policy replaying (blue and red, respectively) with online PEARL and offline MBML.

**Main Results:**   In Figure 3 we compare our offline algorithm with Thompson sampling based methods, and also with an ablation of the reward relabelling method. For Gridworld, the Thompson sampling method is computed exactly, while for the continuous environments, we use online PEARL [28] – a strong baseline that is *not affected by our offline data limitation*, and MBML [22].[9] For these results the uniform initial state distribution was used to collect data. Note that **we significantly outperform Thompson sampling based methods, demonstrating our claim of learning non-trivial exploration from offline data**. These results are further explained qualitatively by observing the exploration behavior of our learned agents. In Figure 1 and in Figure 5, we visualize the trajectories of the trained agents in the Semi-circle and Ant-Semi-circle domains, respectively.[10] An approximately Bayes-optimal behavior is evident: in the first episode, the agents search for the goal along the semi-circle, and in the second episode, the agents maximize reward by moving directly towards the already found goal. Similar behaviors for Gridworld and Escape-Room are reported in Appendix F. In contrast, a Thompson sampling based agent will never display such search behavior, as *it does not plan* to proactively reduce uncertainty. Instead, such an agent will randomly choose an un-visited possible goal at each episode and directly navigate towards it (cf. Figure 1 in [22]). We further emphasize that the approximately Bayes-optimal search behavior is very different from the training data, in which the agents learned to reach specific goals.

Our results also signify the severity of MDP ambiguity, and align with the theory in Sec. 4. **In domains with non-overlapping identifying states (1-3, 7), as expected, performance without policy replaying (RR) is poor, while in domains with overlapping identifying states policy replaying has little effect**. In Figure 7 in the supplementary, we provide further insight into these results, by plotting the belief update during the episode rollout for Semi-circle: the belief starts as uniform on the semi-circle, and narrows in on the target as the agent explores the semi-circle. With reward relabelling ablated, however, we show that the belief does not update correctly, and the agent believes the reward is at the point it first visited on the semi-circle.

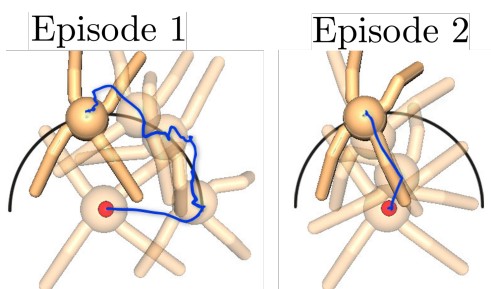

Figure 5: Ant-Semi-circle: trajectories from trained policy on a new goal. In the first episode the ant searches for the goal, and in the second one it directly moves toward the goal it has previously found. This search behavior is different from the goal-reaching behaviors that dominate the data.

---

[9]Since the official implementation of MBML does not support discrete domains nor image observations, we omit this baseline for Gridworld and Reacher-Image. We could not get the BCQ component of MBML to produce reasonable results for Wind with policy replaying, nor for Escape-Room – with or without policy replaying.

[10]Video is provided: `https://youtu.be/6Swg55ZYOU4`

**Data Quality Ablative Study:** To evaluate the dependency of our method on the offline data quality, we report results for the 3 different data collection strategies described above (see supplementary for more details), summarized in Table 1. As expected, data diversity is instrumental to offline training. However, as we qualitatively show in Figure 10 in the supplementary, even on the low-diversity datasets, our agents learned non-trivial exploration strategies that search for the goal. This is especially remarkable for the fixed-distribution dataset, where it is unlikely that any training trajectory traveled along the semi-circle (see supplementary Figure 11).

Table 1: Average return in Ant-Semi-circle for offline data with different initial state distributions: **Uniform** distribution, uniform distribution excluding states on the semi-circle (**Excluding s.c.**), and fixed initial position (**Fixed**).

|  | BOReL | BOReL+CQL |
|---|---|---|
| Uniform | $171.8 \pm 7.0$ | $176.0 \pm 10.2$ |
| Excluding s.c. | $102.8 \pm 32.7$ | $116.6 \pm 19.9$ |
| Fixed | $99.2 \pm 27.4$ | $112.4 \pm 31.3$ |

One may ask whether OMRL presents the same challenge as standard offline RL, and whether recent offline RL advances can mitigate the dependency on data diversity. To investigate this, we also compare our method with a variant that uses CQL [19] – a state-of-the-art offline RL method – to train the critic network of the meta-RL agent. Interestingly, while CQL improved results (Table 1), the effect of data diversity is much more significant. Together with our results on MDP ambiguity, our investigation highlights the particular challenges of the OMRL problem.

## 7    Conclusion and Future Work

We presented the first approximately Bayes-optimal offline meta-RL algorithm, allowing to solve problems where efficient exploration is crucial. The connection between Bayesian RL and meta learning allows to reduce the problem to offline RL on belief-augmented states. However, learning a neural belief update from offline data is prone to MDP ambiguity. We formalized the problem, and proposed a simple data collection protocol that guarantees identifiability. In the particular case of tasks that differ in their rewards, our protocol can be implemented in hindsight, for arbitrarily offline data. Finally, we demonstrated state-of-the-art results on several challenging domains.

An important investigation that we leave to future work is to formalize the connection between task diversity and task generalization. Additionally, it is intriguing whether other techniques can mitigate MDP ambiguity, for example, by designing exploration policies that induce identifiability, or by using domain knowledge to restrict the hypothesis set of possible MDPs in the belief.

## Acknowledgments

This work is partly funded by the Israel Science Foundation (ISF-759/19) and the Open Philanthropy Project Fund, an advised fund of Silicon Valley Community Foundation. The authors would like to thank Luisa Zintgraf for sharing the VariBAD code.

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
