# A   Propositions Proofs

For ease of reading, we copy here the propositions from the main text.

**Proposition 1.** *Let $\tau = s_0, a_0, r_1, s_1 \ldots, r_H, s_H$ denote a random trajectory from a fixed history dependent policy $\pi$, generated according to the following process. First, MDP parameters $\mathcal{R}, \mathcal{P}$ are drawn from the prior $p(\mathcal{R}, \mathcal{P})$. Then, the state trajectory is generated according to $s_0 \sim P_{init}$, $a_t \sim \pi(\cdot|s_0, a_0, r_1, \ldots, s_t)$, $s_{t+1} \sim \mathcal{P}(\cdot|s_t, a_t)$ and $r_{t+1} \sim \mathcal{R}(s_t, a_t)$. Let $b_t$ denote the posterior belief at time $t$, $b_t = P(\mathcal{R}, \mathcal{P}|s_0, a_0, r_1, \ldots, s_t)$. Then*

$$P(s_{t+1}|s_0, a_0, r_1, \ldots, r_t, s_t, a_t) = \mathbb{E}_{\mathcal{R}, \mathcal{P} \sim b_t} \mathcal{P}(s_{t+1}|s_t, a_t), \ and,$$

$$P(r_{t+1}|s_0, a_0, r_1, \ldots, s_t, a_t) = \mathbb{E}_{\mathcal{R}, \mathcal{P} \sim b_t} \mathcal{R}(r_{t+1}|s_t, a_t).$$

*Proof.* For the transitions, we have that,

$$
\begin{aligned}
P(s_{t+1}|s_0, a_0, r_0, \ldots, r_t, s_t, a_t) &= \int P(s_{t+1}, \mathcal{R}, \mathcal{P}|s_0, a_0, r_0, \ldots, r_t, s_t, a_t) d\mathcal{R} d\mathcal{P} \\
&= \int P(s_{t+1}|\mathcal{R}, \mathcal{P}, s_0, a_0, r_0, \ldots, r_t, s_t, a_t) P(\mathcal{R}, \mathcal{P}|s_0, a_0, r_0, \ldots, r_t, s_t, a_t) d\mathcal{R} d\mathcal{P} \\
&= \mathbb{E}_{\mathcal{R}, \mathcal{P}} \left[ P(s_{t+1}|\mathcal{R}, \mathcal{P}, s_0, a_0, r_0, \ldots, r_t, s_t, a_t) \middle| s_0, a_0, r_0, \ldots, r_t, s_t, a_t \right] \\
&= \mathbb{E}_{\mathcal{R}, \mathcal{P}} \left[ \mathcal{P}(s_{t+1}|s_t, a_t) \middle| s_0, a_0, r_0, \ldots, r_t, s_t, a_t \right] \\
&= \mathbb{E}_{\mathcal{R}, \mathcal{P} \sim b_t} \mathcal{P}(s_{t+1}|s_t, a_t).
\end{aligned}
$$

The proof for the rewards proceeds similarly.  $\square$

**Extended Definitions and Proofs for Section 4**

For the proofs of identifiability, we start by elaborating the formal definition of our setting. For simplicity, we assume that the MDPs $\{\mathcal{R}_i, \mathcal{P}_i\}_{i=1}^N$ are defined over finite state-action spaces ($|\mathcal{S}|, |\mathcal{A}| < \infty$). For every $i = 1, \ldots, N$, let $\pi_\beta^i$ be a general stationary, stochastic, history-dependent policy[11]. The initial state distribution $P_{init}$ is the same across all MDPs.[12]

We assume that data is collected from trajectories of length at most $T_{\max}$. This is a convenient assumption that holds in every practical scenario, and allows us to side step issues of defining visitation frequencies when $t \to \infty$.

For some $0 \le t \le T_{\max}$, denote by $P_{\mathcal{P}_i, \pi_\beta^i, t}(s, a) = P_{\mathcal{P}_i, \pi_\beta^i}(s_t = s, a_t = a)$ the probability of visiting the state-action pair $(s, a)$ at time $t$ by running the policy $\pi_\beta^i$ on MDP with transition function $\mathcal{P}_i$ and initial state distribution $P_{init}$. Now, we define:

$$P_{\mathcal{P}_i, \pi_\beta}(s, a) = P_{\mathcal{P}_i, \pi_\beta^i} \left( \bigcup_{t \in \{0, \ldots, T_{\max}\}} \{s_t = s, a_t = a\} \right),$$

that is, $P_{\mathcal{P}_i, \pi_\beta}(s, a)$ is the probability of observing state-action $(s, a)$ in the data from MDP $i$. Similarly, we define

$$P_{\mathcal{P}_i, \mathcal{R}_i, \pi_\beta}(s, a, r, s') = P_{\mathcal{P}_i, \pi_\beta}(s, a) \mathcal{P}_i(s'|s, a) P_{\mathcal{R}_i}(r|s, a),$$

the probability of observing the tuple $(s, a, r, s')$ in the data from MDP $i$.

A trajectory from the replay policy $\pi_r$ in MDP $i$ is generated as follows. Let $x$ be a discrete random variable defined on $1, \ldots, N$ with probability $P_x(\cdot)$ that satisfies $P_x(x = k) > 0$ for every $k = 1, \ldots, N$. First, we draw $x$. Then, we sample a trajectory from MDP $i$ using policy $\pi_\beta^x$.

---

[11]We consider stationary policies for notation simplicity, although similar analysis can be made for non-stationary policies.

[12]The idea of policy replaying can be extended to MDP with different initial state distributions by randomly selecting the state distribution along with the policy. For simplicity, we do not consider this case, although a similar analysis holds for it.

**Proposition 2.** *Consider the setting described in Definition 1. For a pair of MDPs $i$ and $j$, we define the identifying state-action pairs as the state-action pairs that satisfy $\mathcal{R}_i(s,a) \neq \mathcal{R}_j(s,a)$ and/or $\mathcal{P}_i(s'|s,a) \neq \mathcal{P}_j(s'|s,a)$. If for every $i \neq j$ there exists an identifying state-action pair that has positive probability under both $i$ and $j$, i.e., $P_{\mathcal{P}_i,\pi^i_\beta}(s,a), P_{\mathcal{P}_j,\pi^j_\beta}(s,a) > 0$, then the data is identifiable.*

Before we prove Proposition 2, we present the following lemma, which will be used later in the proof.

**Lemma 1.** *Consider a pair of MDPs $(\mathcal{R},\mathcal{P})$ and $(\mathcal{R}',\mathcal{P}')$, and two policies $\pi$ and $\pi'$. If there exists an identifying state-action pair of the MDPs $(\bar{s},\bar{a})$ that has positive probability under both $(\mathcal{P},\pi)$ and $(\mathcal{P}',\pi')$, i.e., $P_{\mathcal{P},\pi}(\bar{s},\bar{a}), P_{\mathcal{P}',\pi'}(\bar{s},\bar{a}) > 0$, then $P_{\mathcal{R},\mathcal{P},\pi}(s,a,r,s') \neq P_{\mathcal{R}',\mathcal{P}',\pi'}(s,a,r,s')$.*

*Proof.* Assume to the contrary that $P_{\mathcal{R},\mathcal{P},\pi}(s,a,r,s') = P_{\mathcal{R}',\mathcal{P}',\pi'}(s,a,r,s')$. Marginalizing over $r$ and $s'$, we obtain:

$$\sum_{r,s'} P_{\mathcal{R},\mathcal{P},\pi}(s,a,r,s') = \sum_{r,s'} P_{\mathcal{R},\mathcal{P}',\pi'}(s,a,r,s')$$

$$P_{\mathcal{P},\pi}(s,a) = P_{\mathcal{P}',\pi'}(s,a), \quad \forall(s,a).$$

Specifically, we have $P_{\mathcal{P},\pi}(\bar{s},\bar{a}) = P_{\mathcal{P}',\pi'}(\bar{s},\bar{a})$. Since $P_{\mathcal{R},\mathcal{P},\pi}(\bar{s},\bar{a},r,s') = P_{\mathcal{R},\mathcal{P}}(r,s'|\bar{s},\bar{a})P_{\mathcal{P},\pi}(\bar{s},\bar{a})$ for every $r$ and $s'$, and $P_{\mathcal{P},\pi}(\bar{s},\bar{a}) = P_{\mathcal{P}',\pi'}(\bar{s},\bar{a}) > 0$, it holds that $P_{\mathcal{R},\mathcal{P}}(r,s'|\bar{s},\bar{a}) = P_{\mathcal{R}',\mathcal{P}'}(r,s'|\bar{s},\bar{a})$ for every $r$ and $s'$. By marginalizing over $s'$ we get that

$$\sum_{s'} P_{\mathcal{R},\mathcal{P}}(r,s'|\bar{s},\bar{a}) = \sum_{s'} P_{\mathcal{R}',\mathcal{P}'}(r,s'|\bar{s},\bar{a})$$

$$P_{\mathcal{R}}(r|\bar{s},\bar{a}) = P_{\mathcal{R}'}(r|\bar{s},\bar{a}).$$

Similarly, by marginalizing over $r$, we get $\mathcal{P}_i(s'|\bar{s},\bar{a}) = \mathcal{P}_j(s'|\bar{s},\bar{a})$. Overall, both reward and transition function do not differ in $(\bar{s},\bar{a})$, which contradicts the fact that $(\bar{s},\bar{a})$ is an identifying state-action pair. $\qquad\square$

We now prove Proposition 2.

*Proof.* Consider some $i \neq j$. Let $(s_{i,j},a_{i,j})$ be an identifying state-action pair that has positive probability under both $i$ and $j$. Assume to the contrary that there exists an MDP $\{\mathcal{R},\mathcal{P}\} \in \mathcal{M}$ and two policies $\pi$ and $\pi'$ such that $P_{\mathcal{R}_i,\mathcal{P}_i,\pi^i_\beta}(s,a,r,s') = P_{\mathcal{R},\mathcal{P},\pi}(s,a,r,s')$ and $P_{\mathcal{R}_j,\mathcal{P}_j,\pi^j_\beta}(s,a,r,s') = P_{\mathcal{R},\mathcal{P},\pi'}(s,a,r,s')$.

Since $(s_{i,j},a_{i,j})$ has positive probability under $(\mathcal{P}_i,\pi^i_\beta)$ and $P_{\mathcal{R}_i,\mathcal{P}_i,\pi^i_\beta}(s,a,r,s') = P_{\mathcal{R},\mathcal{P},\pi}(s,a,r,s')$, then $(s_{i,j},a_{i,j})$ must also have positive probability under $(\mathcal{P},\pi)$ (otherwise, there are $r$ and $s'$ for which $P_{\mathcal{R}_i,\mathcal{P}_i,\pi^i_\beta}(s_{i,j},a_{i,j},r,s') > 0$, while $P_{\mathcal{R},\mathcal{P},\pi}(s_{i,j},a_{i,j},r,s') = 0$). Now, since $(s_{i,j},a_{i,j})$ has positive probability under both $(\mathcal{P}_i,\pi^i_\beta)$ and $(\mathcal{P},\pi)$, and $P_{\mathcal{R}_i,\mathcal{P}_i,\pi^i_\beta}(s,a,r,s') = P_{\mathcal{R},\mathcal{P},\pi}(s,a,r,s')$, according to Lemma 1, it cannot be an identifying state-action pair of $(\mathcal{R}_i,\mathcal{P}_i)$ and $(\mathcal{R},\mathcal{P})$. Therefore, the MDP $\{\mathcal{R},\mathcal{P}\}$ must satisfy $\mathcal{P}(\cdot|s_{i,j},a_{i,j}) = \mathcal{P}_i(\cdot|s_{i,j},a_{i,j})$ and $\mathcal{R}(s_{i,j},a_{i,j}) = \mathcal{R}_i(s_{i,j},a_{i,j})$.

The same argument can be made for $(\mathcal{R}_j,\mathcal{P}_j,\pi^j_\beta)$ and $(\mathcal{R},\mathcal{P},\pi')$, resulting in $\mathcal{P}(\cdot|s_{i,j},a_{i,j}) = \mathcal{P}_j(\cdot|s_{i,j},a_{i,j})$ and $\mathcal{R}(s_{i,j},a_{i,j}) = \mathcal{R}_j(s_{i,j},a_{i,j})$. Overall, we get $\mathcal{P}_i(\cdot|s_{i,j},a_{i,j}) = \mathcal{P}(\cdot|s_{i,j},a_{i,j}) = \mathcal{P}_j(\cdot|s_{i,j},a_{i,j})$ and $\mathcal{R}_i(s_{i,j},a_{i,j}) = \mathcal{R}(s_{i,j},a_{i,j}) = \mathcal{R}_j(s_{i,j},a_{i,j})$, which is a contradiction, as $(s_{i,j},a_{i,j})$ is an identifying state-action pair of MDPs $i$ and $j$. $\qquad\square$

**Proposition 3.** *For every $i \neq j$, denote the set of identifying state-action pairs by $\mathcal{I}_{i,j}$. If for every $i$ and every $j$ exists $(s_{i,j},a_{i,j}) \in \mathcal{I}_{i,j}$ such that $P_{\mathcal{P}_i,\pi^i_\beta}(s_{i,j},a_{i,j}) > 0$, then replacing $\pi^i_\beta$ with $\pi_r$ for all $i$ results in identifiable data.*

*Proof.* Consider some $i \neq j$. We observe that by the construction of $\pi_r$, for every $(s,a)$ pair that satisfies $P_{\mathcal{P}_i,\pi^i_\beta}(s,a) > 0$, we also have $P_{\mathcal{P}_i,\pi_r}(s,a) > 0$. In particular, we have $P_{\mathcal{P}_i,\pi_r}(s_{i,j},a_{i,j}) > 0$.

We will show that either $(s_{i,j}, a_{i,j})$ also has positive probability under $(\mathcal{P}_j, \pi_r)$ or there must exist some other state-action pair that has positive probability under both $(\mathcal{P}_i, \pi_r)$ and $(\mathcal{P}_j, \pi_r)$. This, according to Proposition 2, will result in identifiability of the data.

We define the following sets of state-action pairs:

$$\Sigma_t^i = \{(s,a) : P_{\mathcal{P}_i, \pi_r, t}(s,a) > 0\}, \quad t = 0, 1, \ldots, T_{\max},$$

$$\Sigma_t^{i,j} = \{(s,a) : P_{\mathcal{P}_i, \pi_r, t}(s,a) = P_{\mathcal{P}_j, \pi_r, t}(s,a) > 0\}, \quad t = 0, 1, \ldots, T_{\max}.$$

Note that $\Sigma_0^i = \Sigma_0^{i,j}$, as the initial state distribution $P_{init}$ and $\pi_r$ are fixed across all MDPs.

First, consider the case where for every $t = 0, 1, \ldots, T_{\max}$ we have $\Sigma_t^i = \Sigma_t^{i,j}$. Given that $(s_{i,j}, a_{i,j})$ has positive probability under $(\mathcal{P}_i, \pi_r)$, there exists some $t$ for which $(s_{i,j}, a_{i,j}) \in \mathcal{I}_{i,j} \cap \Sigma_t^i$. Since $\Sigma_t^i = \Sigma_t^{i,j}$, we have $(s_{i,j}, a_{i,j}) \in \mathcal{I}_{i,j} \cap \Sigma_t^{i,j}$, which means $(s_{i,j}, a_{i,j})$ also has positive probability under $(\mathcal{P}_j, \pi_r)$.

Next, consider the case where there exists some $t \in \{1, \ldots, T_{\max}\}$ for which $\Sigma_t^i \neq \Sigma_t^{i,j}$ and let $\hat{t} = \min\{t : \Sigma_t^i \neq \Sigma_t^{i,j}\}$. Note that $\hat{t} > 0$, since we have already shown that $\Sigma_0^i = \Sigma_0^{i,j}$. Thus, for every $t < \hat{t}$ we have $\Sigma_t^i = \Sigma_t^{i,j} = \Sigma_t^j$, and for $\hat{t}$ it holds that $P_{\mathcal{P}_i, \pi_r, \hat{t}}(s,a) \neq P_{\mathcal{P}_j, \pi_r, \hat{t}}(s,a)$. If there exits a $t' < \hat{t} - 1$ and $(s,a) \in \Sigma_{t'}^i$ such that $\mathcal{P}_i(\cdot|s,a) \neq \mathcal{P}_j(\cdot|s,a)$, then we are done as $\Sigma_{t'}^i = \Sigma_{t'}^{i,j}$, which means that $(s,a)$ is an identifying state-action pair that has positive probability under both $(\mathcal{P}_i, \pi_r)$ and $(\mathcal{P}_j, \pi_r)$. Therefore, consider the case where for every $t < \hat{t} - 1$ and every $(s,a) \in \Sigma_t^i$ we have $\mathcal{P}_i(\cdot|s,a) = \mathcal{P}_j(\cdot|s,a)$. We will show that there exists $(s,a) \in \Sigma_{\hat{t}-1}^i$ such that $\mathcal{P}_i(\cdot|s,a) \neq \mathcal{P}_j(\cdot|s,a)$.

Assume to the contrary that for every $(s,a) \in \Sigma_{\hat{t}-1}^i$ we have $\mathcal{P}_i(\cdot|s,a) = \mathcal{P}_j(\cdot|s,a)$, i.e., the transition function is also equivalent for $t = \hat{t} - 1$. Let $h_{\hat{t}} = (x, s_0, a_0, \ldots, s_{\hat{t}}, a_{\hat{t}})$ be the state-action history up to time $\hat{t}$, including the random variable $x$ that was used to choose a policy. We next consider the probability of observing a history under $(\mathcal{P}_i, \pi_r)$,

$$P_{\mathcal{P}_i, \pi_r}(h_{\hat{t}}) = P_{init}(s_0) P_x(x) \pi_r(a_0|x, s_0) P_{\mathcal{P}_i, \pi_r}(s_1|x, s_0, a_0) \pi_r(a_1|x, s_0, a_0, s_1) \cdots$$
$$\cdots P_{\mathcal{P}_i, \pi_r}(s_{\hat{t}}|x, s_0, a_0, \ldots, s_{\hat{t}-1}, a_{\hat{t}-1}) \pi_r(a_{\hat{t}}|x, s_0, a_0, \ldots, s_{\hat{t}})$$
$$= P_{init}(s_0) P_x(x) \pi_r(a_0|x, s_0) \prod_{t=1}^{\hat{t}} \mathcal{P}_i(s_t|s_{t-1}, a_{t-1}) \pi_r(a_t|x, s_0, a_0, \ldots, s_t),$$

where the last equality holds according to the Markov property, $P_{\mathcal{P}_i, \pi_r}(s_t|s_0, a_0, \ldots, s_{t-1}, a_{t-1}) = \mathcal{P}_i(s_t|s_{t-1}, a_{t-1})$. Since $\pi_r(a_t|x, s_0, a_0, \ldots, s_t)$ is the same replaying policy for all MDPs, and for every $t \leq \hat{t} - 1$ and $(s,a) \in \Sigma_t^i$ we have $\mathcal{P}_i(\cdot|s,a) = \mathcal{P}_j(\cdot|s,a)$, then $P_{\mathcal{P}_i, \pi_r}(h_{\hat{t}}) = P_{\mathcal{P}_j, \pi_r}(h_{\hat{t}})$. By marginalizing over $x, s_0, a_0, \ldots, s_{\hat{t}-1}, a_{\hat{t}-1}$ we obtain:

$$\sum_{x, s_0, a_0, \ldots, s_{\hat{t}-1}, a_{\hat{t}-1}} P_{\mathcal{P}_i, \pi_r}(x, s_0, a_0, \ldots, s_{\hat{t}}, a_{\hat{t}}) = \sum_{x, s_0, a_0, \ldots, s_{\hat{t}-1}, a_{\hat{t}-1}} P_{\mathcal{P}_j, \pi_r}(x, s_0, a_0, \ldots, s_{\hat{t}}, a_{\hat{t}})$$

$$P_{\mathcal{P}_i, \pi_r}(s_{\hat{t}}, a_{\hat{t}}) = P_{\mathcal{P}_j, \pi_r}(s_{\hat{t}}, a_{\hat{t}}),$$

which means that $\Sigma_{\hat{t}}^i = \Sigma_{\hat{t}}^{i,j}$, which contradicts the definition of $\hat{t}$.

$\square$

# B  BOReL Pseudo-Code

---

**Algorithm 1** BOReL

---

**Input:** A set of MDPs $\{\mathcal{R}_i, \mathcal{P}_i\}_{i=1}^N \sim p(\mathcal{R}, \mathcal{P})$; $k$ – # of consecutive episodes that comprise a trajectory for VAE training.

*Phase 1: Data Collection*

**for** $i = 1, \ldots, N$ **do**

    Train standard RL agent (e.g., DQN, SAC) to solve $\{\mathcal{R}_i, \mathcal{P}_i\}$

    Save complete training to buffer $\mathcal{D}_i = \{\tau_j^i\}_j$, where $\tau_j^i$ is an episode of length $H$.

**end for**

*Phase 2: Neural Belief Training*

**for** $i = 1, \ldots, N$ **do**

    Concatenate every $k$ consecutive episodes from $\mathcal{D}_i$ to form non-overlapping trajectories of length $H^+ = k \times H$. Denote them by $\mathcal{D}_i^+ = \{\tau_j^{i^+}\}_j$.

    **for** trajectory $\tau_\ell^{i^+}$ in $\mathcal{D}_i^+$ **do**

        **if** policy replaying **then**

            Uniformly draw a trained policy $\tilde{\pi}_\ell \sim \{\pi_\beta^j\}_{j=1}^N$.

            Collect $k/2$ episodes $\{\tilde{\tau}_j\}_{j=1}^{k/2}$ by running $\tilde{\pi}_\ell$ on $\{\mathcal{R}_i, \mathcal{P}_i\}$.

            *Replace* the first or last (uniformly chosen) $k/2$ episodes of $\tau_\ell^{i^+}$ with $\{\tilde{\tau}_j\}_{j=1}^{k/2}$.

        **else if** reward relabelling **then**

            Uniformly draw $j \sim \{1, \ldots, N\}$.

            *Replace* the rewards $r_{t+1}^{i,\ell}(s_t^{i,\ell}, a_t^{i,\ell})$ in the first or last (uniformly chosen) $k/2$ episodes of $\tau_\ell^{i^+}$ with rewards $\hat{r}_{t+1}^{i,\ell} = \mathcal{R}_j(s_t^{i,\ell}, a_t^{i,\ell})$.

        **end if**

    **end for**

**end for**

Train VAE according to (4) using trajectories from $\mathcal{D}^+ := \{\mathcal{D}_1^+, \ldots, \mathcal{D}_N^+\}$.

*Phase 3: State Relabelling*

**for** trajectory $\tau^+$ *in* $\mathcal{D}^+$ **do**

    **for** $t = 1, \ldots, H^+$ **do**

        Pass $\tau_{:t}^+$ through encoder to obtain $b_t = \mu(\tau_{:t}^+), \Sigma(\tau_{:t}^+)$

        *Replace* $s_t$ in trajectory with $s_t^+ = (s_t, b_t)$.

    **end for**

**end for**

*Phase 4: Offline Meta-RL Training*

Train off-policy RL agent (e.g., DQN, SAC) using the offline data obtained from Phase 3.

---

# C  VAE Training Objective

For completeness, we follow [36] and outline the full training objective of the VAE. Consider the approximate posterior $q_\phi(m|h_{:t})$ conditioned on the history up to time $t$. In this case, the ELBO can be derived as follows:

$$
\begin{aligned}
\log P(s_0, r_1, s_1 \ldots, s_H | a_0, \ldots, a_{H-1}) &= \log \int P(s_0, r_1, s_1 \ldots, s_H, m | a_0, \ldots, a_{H-1}) dm \\
&= \log \mathbb{E}_{m \sim q_\phi(\cdot|h_{:t})} \left[ \frac{P(s_0, r_1, s_1 \ldots, s_H, m | a_0, \ldots, a_{H-1})}{q_\phi(m|h_{:t})} \right] \\
&\geq \mathbb{E}_{m \sim q_\phi(\cdot|h_{:t})} \left[ \log p_\theta(s_0, r_1, s_1 \ldots, s_H | m, a_0, \ldots, a_{H-1}) \right. \\
&\quad \left. + \log p_\theta(m) - \log q_\phi(m|h_{:t}) \right] \\
&= \mathbb{E}_{m \sim q_\phi(\cdot|h_{:t})} \left[ \log p_\theta(s_0, r_1, s_1 \ldots, s_H | m, a_0, \ldots, a_{H-1}) \right] \\
&\quad - D_{KL}(q_\phi(m|h_{:t}) || p_\theta(m)) \\
&= ELBO_t(\theta, \phi).
\end{aligned}
$$

The prior $p_\theta(m)$ is set to be the previous posterior $q_\phi(m|h_{:t-1})$, with initial prior chosen to be standard normal $p_\theta(m) = \mathcal{N}(0, I)$. The decoder $p_\theta(s_0, r_1, s_1 \ldots, s_H | m, a_0, \ldots, a_{H-1})$ factorizes to reward and next state models $p_\theta(s'|s, a, m)$ and $p_\theta(r|s, a, m)$, according to:

$$\log p_\theta(s_0, r_1, s_1 \ldots, s_H | m, a_0, \ldots, a_{H-1}) = \log p(s_0|m)$$
$$+ \sum_{t=0}^{H-1} \left[ \log p_\theta(s_{t+1}|s_t, a_t, m) + \log p_\theta(r_{t+1}|s_t, a_t, m) \right].$$

The overall training objective of the VAE is to maximize the sum of ELBO terms for different time steps,

$$\max_{\theta, \phi} \sum_{t=0}^{H} ELBO_t(\theta, \phi). \tag{4}$$

## D  Environments Description

In this section we describe the details of the domains we experimented with.

**Gridworld:**  A $5 \times 5$ gridworld environment as in [36]. The task distribution is defined by the location of a goal, which is unobserved and can be anywhere but around the starting state at the bottom-left cell. For each task, the agent receives a reward of $-0.1$ on non-goal cells and $+1$ at the goal, i.e.,

$$r_t = \begin{cases} 1, & s_t = g \\ -0.1, & \text{else}, \end{cases}$$

where $s_t$ is the current cell and $g$ is the goal cell.
Similarly to [36], the horizon for this domain is set to 15 and we aggregate $k = 4$ consecutive episodes to form a trajectory of length 60.

**Semi-circle:**  A continuous 2D environment as in Figure 1, where the agent must navigate to an unknown goal, randomly chosen on a semi-circle of radius 1 [28]. For each task, the agent receives a reward of $+1$ if it is within a small radius $r = 0.2$ of the goal, and 0 otherwise,

$$r_t = \begin{cases} 1, & \|x_t - x_{\text{goal}}\|_2 \le r \\ 0, & \text{else}, \end{cases}$$

where $x_t$ is the current 2D location. Action space is 2-dimensional and bounded: $[-0.1, 0.1]^2$.
We set the horizon to 60 and aggregate $k = 2$ consecutive episodes to form a trajectory of length 120.

**MuJoCo:**

1. **Half-Cheetah-Vel:** In this environment, a half-cheetah agent must run at a fixed target velocity. Following recent works in meta-RL [9, 28, 36], we consider velocities drawn uniformly between $0.0$ and $3.0$. The reward in this environment is given by

   $$r_t = -|v_t - v_{\text{goal}}| - 0.05 \cdot \|a_t\|_2^2$$

   where $v_t$ is the current velocity, and $a_t$ is the current action. The horizon is set to 200 and we aggregate $k = 2$ consecutive episodes.

2. **Ant-Semi-circle:** In this environment, an ant needs to navigate to an unknown goal, randomly chosen on a semi-circle, similarly to the Semi-circle task above.

   When collecting data for this domain, we found that the standard SAC algorithm [17] was not able to solve the task effectively due to the sparse reward (which is described later), and did not produce trajectories that reached the goal. We thus modified the reward **only during data collection** to be dense, and inversely proportional to the distance from the goal,

   $$r_t^{\text{dense}} = -\|x_t - x_{\text{goal}}\|_1 - 0.1 \cdot \|a_t\|_2^2$$

   where $x_t$ is the current 2D location and $a_t$ is the current action. After collecting the data trajectories, we replaced all the dense rewards in the data with the sparse rewards that are given by

   $$r_t^{\text{sparse}} = -0.1 \cdot \|a_t\|_2^2 + \begin{cases} 1, & \|x_t - x_{\text{goal}}\|_2 \le 0.2 \\ 0, & \text{else}. \end{cases}$$

We note that [28] use a similar approach to cope with sparse rewards in the online setting. The horizon is set to 200 and we aggregate $k = 2$ consecutive episodes.

3. **Reacher-Image:** In this environment, a two-link planar robot needs to reach an unknown goal, randomly chosen on a quarter circle. The robot receives dense reward which is given by

$$r_t = -\|x_t - x_{\text{goal}}\|_2$$

where $x_t$ is the location of the robot's end effector. The agent observes single-channel images of size $64 \times 64$ of the environment (see Figure 6b). The horizon is set to 100 and we aggregate $k = 2$ consecutive episodes.

**Wind:** A continuous 2D domain with varying transitions, where the agent must navigate to a fixed (unknown) goal within a distance of $D = 1$ from its initial state (the goal location is the same for all tasks). Similarly to Semi-circle, the agent receives a reward of $+1$ if it is within a radius $r = 0.2$ of the goal, and 0 otherwise,

$$r_t = \begin{cases} 1, & \|s_t - s_{\text{goal}}\|_2 \leq r \\ 0, & \text{else.} \end{cases}$$

For each task in this domain, the agent is experiencing a different 'wind', which results in a shift in the transitions, such that when taking action $a_t \in [-0.1, 0.1]^2$ from state $s_t$ in MDP $\mathcal{M}$, the agent transitions to a new state $s_{t+1}$, which is given by

$$s_{t+1} = s_t + a_t + w_{\mathcal{M}},$$

where $w_{\mathcal{M}}$ is a task-specific wind, which is randomly drawn for each task from a uniform distribution over $[-0.05, 0.05]^2$. To navigate correctly to the goal and stay there, the agent must take actions that cancel the wind effect.
We set the horizon to 25 and evaluate the performance in terms of average return within the **first** episode of interaction on test tasks ($k = 1$).

**Escape-Room:** A continuous 2D domain where the agent must navigate outside a circular room of radius $R = 1$ through an opening, whose location is unknown. For all tasks, the central angle of the opening is $\pi/8$. The tasks differ by the location of the opening – the center point of the opening is sampled uniformly from $[0, \pi]$. The reward function is sparse, task-independent, and given by

$$r_t = \begin{cases} 1, & \|s_t\|_2 > R \\ 0, & \text{else.} \end{cases}$$

The transition function, however, is task-dependent and given by

$$s_{t+1} = \begin{cases} \frac{s_t + a_t}{\|s_t + a_t\|_2}, & \text{if } \textit{intersection occurs} \\ s_t + a_t, & \text{else,} \end{cases}$$

where *intersection occurs* means that the line that connects $s_t$ and $s_t + a_t$ and the wall of the circular room intersects. To solve a task, the agent must search for the opening by moving along the wall until he finds it.
We set the horizon to 60 and aggregate $k = 2$ consecutive episodes to a form a trajectory of length 120.

## E   Experimental Details

In this section we outline our training process and hyperparameters. Additional details and code can be found at `https://github.com/AnonNeurIPS2021/BOReL`. For training, we used a server with 4 NVIDIA RTX 2080 GPUs. Note that collecting offline data for each domain requires to first successfully train a large number of 'standard' RL agents, which can be demanding; we will make our data available publicly.

For the discrete Gridworld domain we used DQN [25] with soft target network updates, as proposed by [23], which was shown to improve the stability of learning. For the rest of the continuous domains, we used SAC [17] with the architectures of the actor and critic chosen similarly, and with a fixed entropy coefficient. For both DQN and SAC, we set the soft target update parameter to 0.005.

In our experiments we average performance over 3 random seeds and present the mean and standard deviation.

Our offline training procedure is comprised of 3 separate training steps. First is the training of the data collection RL agents. Each agent is trained on a different task from the task distribution.

For all domains but Reacher-Image, we used a similar architecture of 2 fully-connected (FC) hidden layers of size that depends on the domain with ReLU activations, and set the batch size to 256.

For Reacher-Image, we used data augmentation techniques as suggested by Laskin et al. (2020). Specifically, we used random translations and cropping. Then, we pass the observation through a convolutional neural network (CNN) with 4 hidden layers followed by 2 FC hidden layers.

The rest of the hyperparameters used for training the data collection RL agents are summarized in the following table:

| | Gridworld | Semi-circle | Cheetah & Ant | Reacher | Wind | Escape-Room |
|---|---|---|---|---|---|---|
| **Num. train tasks** | 21 | 80 | 100 & 80 | 50 | 40 | 60 |
| Hidden layers size | 16 | 32 | 128 | 1024 | 64 | 128 |
| Num. iterations | 200 | 300 | 1000 | 50 | 300 | 50 |
| RL updates per iter. | 500 | 500 | 2000 | 500 | 500 | 500 |
| Exploration/ entropy coeff. | $\epsilon$-greedy, annealing from 1 to 0.1 over 100 iterations | 0.01 | 0.2 | 0.05 | 0.01 | 0.01 |
| Collected ep. per iter. | 5 | 2 | 2 | 1 | 2 | 2 |
| Learning rate/s | $3 \cdot 10^{-4}$ | $3 \cdot 10^{-4}$ | $3 \cdot 10^{-4}$ | $1 \cdot 10^{-3}$ | $3 \cdot 10^{-4}$ | $3 \cdot 10^{-4}$ |
| Discount factor ($\gamma$) | 0.99 | 0.9 | 0.99 | 0.99 | 0.9 | 0.9 |

The second training step is the VAE training after optionally applying reward relabelling/policy replaying to the collected data.

The VAE consists of a recurrent encoder, which at time step $t$ takes as input the tuple $(a_t, r_{t+1}, s_{t+1})$. The state and reward are passed each through a different fully-connected (FC) layer (preceded by a CNN feature-extractor in Reacher-Image). The state FC layer is of size 32 and the reward FC layer is of size 8 for the Gridworld and 16 for the rest of the domains, all with ReLU activations. For all environments but Gridworld, we also pass the action through a FC layer of size 16 with ReLU. Then, the state and reward layers' outputs are concatenated along with the action (or with the output of the action layer) and passed to a GRU of size 64/128 (Gridworld/other domains). The GRU outputs the Gaussian parameters $\mu(h_{:t}), \Sigma(h_{:t})$ of the latent vector $m$, whose dimensionality is 5 in all our experiments.

For all reward-varying domains (all but Wind/Escape-Room), we only train reward-decoder (Similarly to Zintgraf et al. [36]). For Wind and Escape-room we also train transition decoder. In all domains, the decoder network/s are comprised of 2 FC layers, each of size 32.

The VAE is trained to optimize Equation (4), but similarly to [36], we weight the KL term in each of the ELBO terms with some parameter $\beta$, which is not necessarily 1. In our experiments we used $\beta = 0.05$.

After the VAE is trained, we apply state relabelling to the data collected by the RL agents, to create a large offline dataset that effectively comes from the BAMDP. Then, we train an off-policy RL algorithm, which is our meta-RL agent, using the offline data.

For the offline meta-RL agents training, we used similar hyperparameters to those used for the data collection RL agents training. For some of the domains, we enlarge the size of the hidden layers.

# F   Learned Belief and Policy Visualizations

In this section we visualize the learned belief states, in order to get more insight into the decision making process of the agent during interaction. We also visualize trajectories of trained agents in different domains.

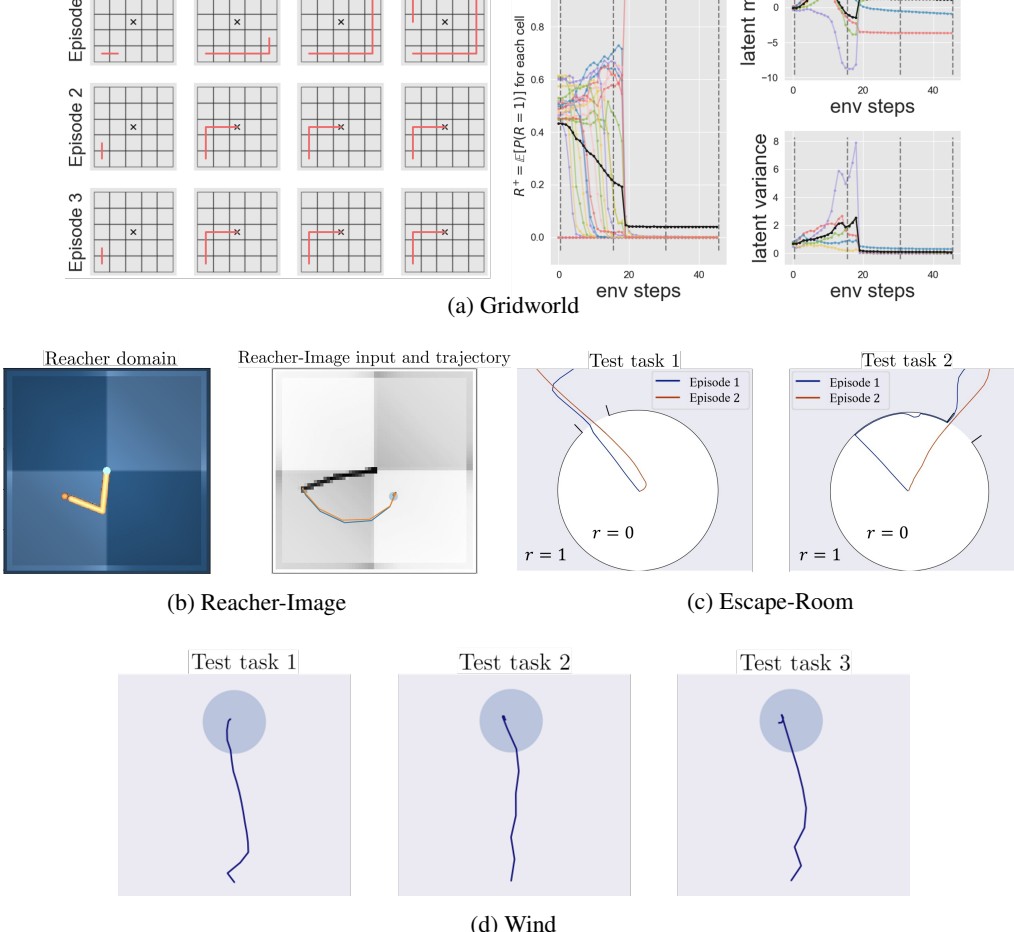

Figure 6: Interaction of trained agents with evaluated domains. In (a) we show interaction with Gridworld, including belief update throughout interaction (for more details, see [36]). In (b), (c) and (d) we show typical behavior of trained agents interacting with Reacher-Image, Escape-Room and Wind, respectively.

In Figure 6a, we visualize the interaction of a trained agent with the Gridworld environment, exactly as visualized in Figure 3 at [36]. The agent reduces its uncertainty by effectively searching the goal. After the goal is found, the agent stops and in subsequent episodes it directly moves toward it.

The Reacher domain is visualized in Figure 6b. In the left side, an RGB image of the domain is presented. In the right side, we present the input image to the agent (which consists of a single-channel and has lower resolution) along with successful trajectories that reaches a goal from the test set.

In Figure 6c we show the typical behavior of a trained agent in Escape-Room domain. Note that in 'Test task 1' the agent finds the opening without colliding with the walls of the room and in the second episode the agent follows a similar trajectory that leads to reward. On the other hand, in 'Test task 2' the agent collides with the wall in the first episode, and then it effectively searches for the opening. After he finds it, in the second episode he directly escapes the room.

In Figure 6d we visualize trajectories of a trained agent on different test tasks in Wind domain. As can be seen, after several steps in the environment, our agent learns to adapt to the varying wind, and travels to the goal in a straight line. PEARL, on the other hand, only adapts after the first episode, and therefore obtains worse results (see Figure 4). We believe it is possible to improve PEARL to update its posterior after every step, and in this case the improved PEARL will obtain similar performance as our method in Wind. However, this will not work in the sparse domains described in the main

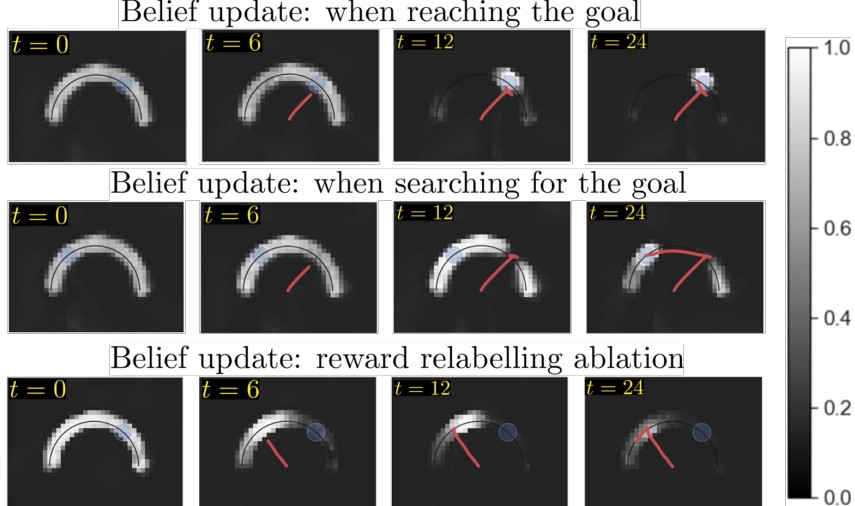

Figure 7: Semi-circle belief visualization. The plots show the reward belief over the 2-dimensional state space (obtained from the VAE) at different stages of interacting with the system. The red line marks the agent trajectory, and the light blue circle marks the true reward location. **Top:** Once the agent finds the true goal, it reduces the belief over other possible goals from the task distribution. **Middle:** As long as the agent doesn't find the goal, it explores efficiently, reducing the uncertainty until the goal is found. **Bottom:** Without reward relabelling, the agent doesn't learn to differentiate between different MDPs, and therefore fails to identify the goal.

text, where the Bayes adaptive exploration has an inherent advantage over Thompson sampling. We emphasize that in Wind, MDP ambiguity is not a concern, since the data from all agents is largely centered on the line between the agent's initial position and the goal. Thus, the effect of the wind on these states can uniquely be identified in each task.

In Figure 7, we plot the reward belief (obtained from the VAE decoder) at different steps during the agent's interaction in the Semi-circle domain. Note how the belief starts as uniform over the semi-circle, and narrows in on the target as more evidence is collected. Also note that without reward relabelling, the agent fails to find the goal. In this instance of the MDP ambiguity problem, the training data for the meta-RL agent consists of trajectories that mostly reach the goal, and as a result, the agent believes that the reward is located at the first point it reaches on the semi-circle.

## G   Data Quality Ablation

In our data quality ablative study, we consider the Ant-Semi-circle domain for which we modify the initial state distribution during the data collection phase. The initial state distributions we consider are visualized in Figure 8: Uniform distribution, uniform excluding states on the semi-circle, and fixed initial position.

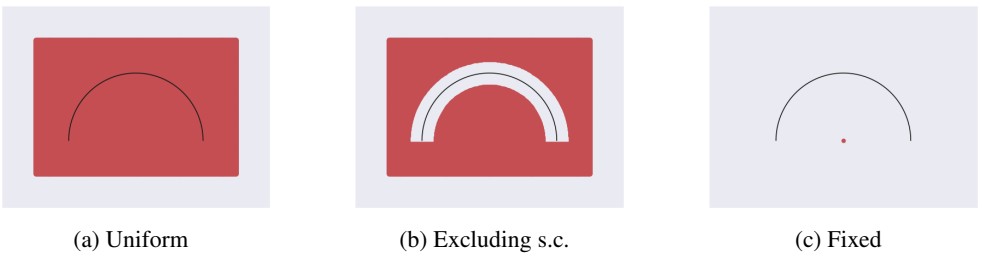

|     (a) Uniform     |     (b) Excluding s.c.     |     (c) Fixed     |

Figure 8: Initial state distributions. Red locations indicate non-zero sampling probability.

Figure 9 shows the learning curves for the results presented in Table 1. For completeness, we add the learning curve for the uniform distribution which is also presented in Figure 3.

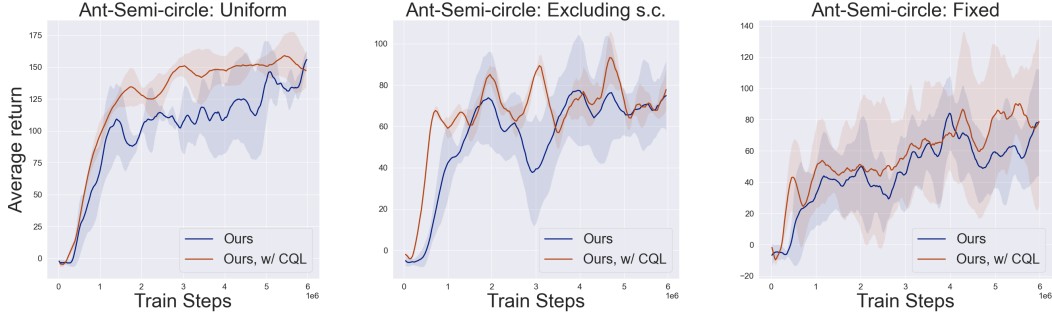

Figure 9: Learning curves for the results presented in Table 1. In blue is our method and in red is our method with critic network trained according to the CQL objective [19]. **Left:** Uniform initial state distribution. **Middle:** Uniform distribution, excluding states over the semi-circle. **Right:** Initial state is fixed.

We also visualize trajectories of trained agents for the 3 different cases as well as for MBML [22] (trained with the uniform distribution data) and PEARL [28], in Figure 10.

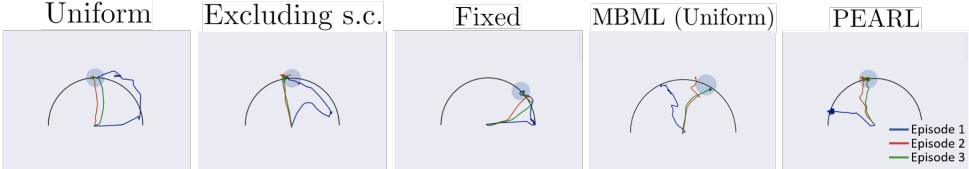

Figure 10: Ant-Semi-circle: trajectories of our trained agents for different offline datasets, of trained MBML agent on the uniform dataset, and PEARL (trained online).

Note that even for the fixed-distribution dataset, our agent learns to search for the goal. To further emphasize why this result is remarkable, in Figure 11 we plot RL agent trajectories, i.e., our offline data, during training with the fixed initial state distribution. Note the significant difference from a Bayes-optimal exploration behavior.

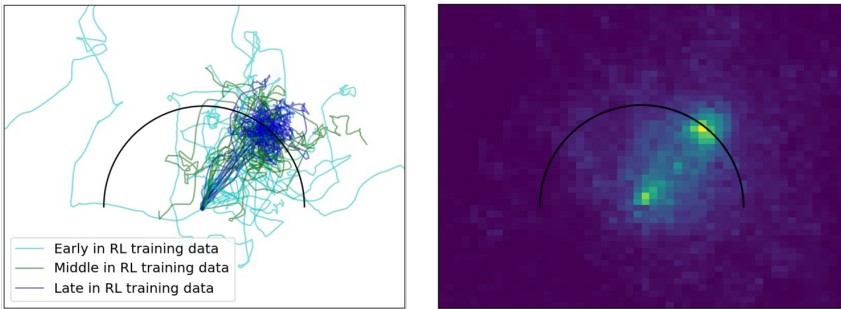

Figure 11: RL agent trajectories during data collection with fixed initial state distribution. **Left:** Trajectories from the beginning (cyan), middle (green), and towards the end of training (blue). **Right:** Heat map corresponding to agent state visitations.

## H   Online Setting Performance

Our method can also be applied to the online setting, in which online data collection is allowed. In this case, it is simply a modification of VariBAD, where the policy gradient optimization is replaced with an off-policy RL algorithm. Since MDP ambiguity does not concern online meta-RL, we did not use reward relabelling in this setting. As shown in Figure 12, by exploiting the efficiency of off-policy RL, our method significantly improves sample-efficiency, without sacrificing final performance.

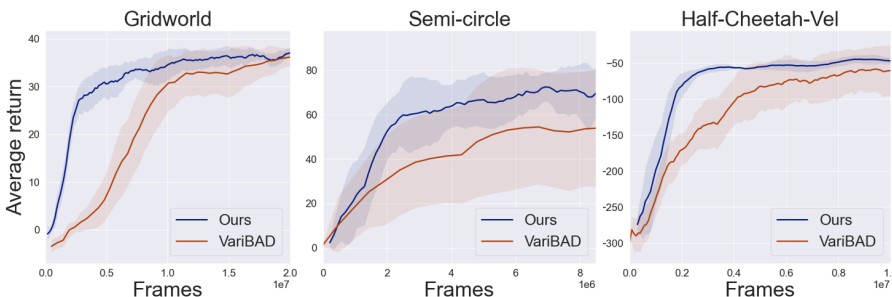

Figure 12: Online performance comparison. The off-policy optimization significantly improved VariBAD performance.

When comparing Figure 12 and Figure 3, the reader may notice that the online algorithm's final performance outperforms the final performance in the offline setting. We emphasize that this phenomenon largely depends on the quality of the offline data, and not on the algorithm itself.

The hyperparameters used in the online setting are as follows:

|  | **Gridworld (DQN)** | **Semi-circle (SAC)** | **Cheetah-Vel (SAC)** |
|---|---|---|---|
| **RL parameters** | | | |
| Architecture/s | 2 FC layers of size 100. | 2 FC layers of size 128. | 3 FC layers of size 128. |
| Num. updates per iter. | 250 | 1000 | 2000 |
| Exploration/entropy coeff. | $\epsilon$-greedy, linear annealing from 1 to 0.1 over 1000 iterations. | 0.01 | 0.2 |
| Collected episodes per iter. | 25 | 25 | 25 |
| Learning rate/s | $7 \cdot 10^{-5}$ | $7 \cdot 10^{-5}$ | $3 \cdot 10^{-4}$ |
| Discount factor ($\gamma$) | 0.99 | 0.9 | 0.99 |
| **VAE parameters** | | | |
| Encoder architecture | state/reward FC layer of size 32/8. GRU of size 64. | state/reward FC layer of size 32/8. GRU of size 128. | state/action/reward FC layer of size 32/16/16. GRU of size 128. |
| Reward decoder architecture | 2 FC layers of size 32. | 2 FC layers of sizes 64 and 32. | 2 FC layers of sizes 64 and 32. |
| Num. updates per iter. | 20 | 25 | 20 |
| Learning rate | $3 \cdot 10^{-4}$ | $10^{-3}$ | $3 \cdot 10^{-4}$ |
| Weight of KL term ($\beta$) | 1.0 | 0.1 | 1.0 |

# I  Additional Results

## I.1  Performance vs. Adaptation Episodes

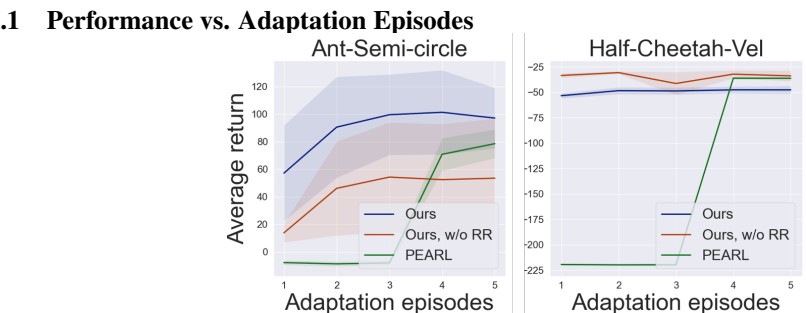

Figure 13: Adaptation performance. Our method outperforms PEARL, collecting high rewards within the first adaptation episodes.

In this part, we present the average reward per-episode as a function of the number of adaptation episodes at the environment. Figure 13 shows the performance for the Ant-Semi-circle and Half-Cheetah-Vel domains. Note that within the first few episodes, PEARL does not collect high rewards due to the Thompson sampling-based nature of the algorithm. Our method, on the other hand, efficiently explores new tasks and is able to collect rewards within the first episodes of interaction.

## I.2 PEARL Learning Curves

We present the training curves of PEARL in Figure 14. Note that since PEARL is an online algorithm, the $x$-axis represents the number of environment interactions.

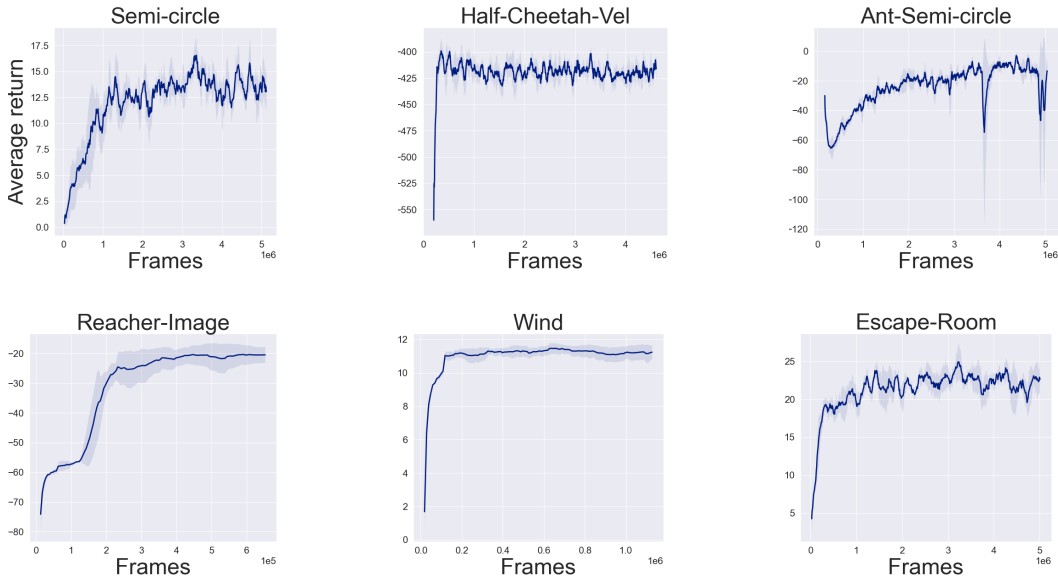

Figure 14: Learning curves for online PEARL training.

## I.3 MBML Learning Curves

We present the training curves of MBML [22] in Figure 15.

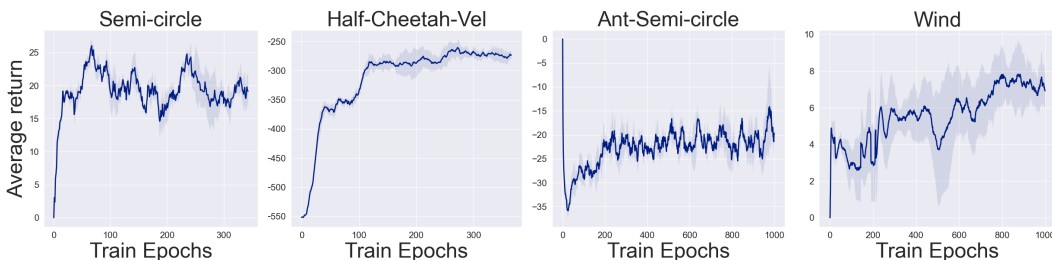

Figure 15: Learning curves for MBML training.