# OpenReview forum: "Offline Meta Reinforcement Learning -- Identifiability Challenges and Effective Data Collection Strategies"
_NeurIPS.cc/2021/Conference — NeurIPS 2021 Poster_

### Official Review · Reviewer_YKZ2 · 2021-07-10

**Rating:** 6
**Confidence:** 4

**Summary:**

The paper addresses the problem of learning Bayes-Optimal Exploration strategies from offline data.
The authors discuss MDP ambiguity problems that may arise in this context and propose BoRel an adaptation
of the online, on-policy VariBad algorithm that addresses these issues via policy replaying and reward relabeling.
Finally, the algorithm is empirically assessed on several baselines and compared to a number of existing (online) methods.

**Ethical Concerns:**

-

**Limitations And Societal Impact:**

-

**Main Review:**

The paper studies a highly relevant problem setting and is technically sound. It is very clearly structured and well-written. The discussion of MDP ambiguity is novel and highlights an important avenue for future research. However, several remaining weaknesses are listed in the following:

1. The adaptation of VariBad for off-policy learning seems to be of limited novelty. The authors claim that VariBad makes no use of the BAMDP parameters. However, the VAE component (approximately) learns to compute a sufficient statistic of the interaction history which the policy network is conditioned on. This can be interpreted as a posterior over the BAMDP parameters and thus the VariBad policy network acts on hyperstates (which is known to be enough to enable computation of the optimal policy).
Switching out the on-policy component with an off-policy one seems trivial as does the re-computation of belief posteriors once the inference VAE is ``  `  fully'' trained.

2. On the topic of training the VAE in the offline setting. The trajectory distribution induced by the final policy is likely to differ significantly from the one induced during data collection. Unless I am mistaken, this would be equivalent to changing the data distribution between training and test time in a ``standard'' supervised context - something that can lead to rather arbitrary behaviour. In light of this, I would be curious as to how a reasonably robust belief computation can be ensured?

3. I find the proposed policy replay strategy somewhat problematic in the context of the offline meta-RL setting. The authors even note that data generally might not be collected specifically for the meta-RL task. Consequently, I do not belief that the agent should be able to prescribe data collection policies.

4. The proposed reward relabeling assumes access to the reward functions of all example tasks which is somewhat limiting. Additionally, it seems to be strongly related to other data augmentation techniques e.g. Hindsight Experience Replay which should at least be mentioned, if not discussed in more detail.

5. While the issue of MDP ambiguity is clearly described and well-illustrated through examples, its consequences in practice remain somewhat unclear. Are there any results regarding the best possible performance one can hope to achieve in its presence? What happens when we cannot make the additional assumptions required for policy replay and reward relabeling?

6. The authors compare to online PEARL claiming it to be a strong baseline, however, other meta-RL algorithm have achieved higher performances (e.g. VariBad itself). Additionally, since PEARL is an online algorithm it needs to both curate its own data (which in BoRel is already done by the expert policies, policy replay and reward relabeling) as well as extract the optimal meta-policy from that data. This implies that PEARL is solving a harder problem and the comparison is somewhat unfair.

**Time Spent Reviewing:**

6

---

> ### Author Response · Authors · 2021-08-09
> **Response to Reviewer YKZ2**
>
> We thank the reviewer for his time and comments, and address the points raised.
>
> 1. This is a delicate point: in the BAMDP framework, the policy conditioned on hyperstates needs to optimize the BAMDP reward function, $R^+$. This is not the case in VariBAD – they use trajectories with rewards sampled from $R$ (MDP-specific). Our theoretical contribution here, as stated in Proposition 1, is that sampling MDP parameters from the prior and then collecting trajectories from the sampled MDP induces a distribution over rewards and transitions that is equivalent (in posterior expectation!) to $R^+$ and $P^+$. This is enough to justify running VariBAD off-policy as we do.
> To clarify, in standard off-policy RL, one could generate $(s,a,r,s’)$ data tuples arbitrarily, so long as $s’\sim P(s’|s,a)$ and $r\sim R(s,a)$. In the BAMDP setting, we could similarly generate ($s^{+}$,a,$r^{+}$,$s^{+}{'}$) data tuples arbitrarily (not as part of a trajectory), but in this case we would not be able to replace $r^{+}$ with some $r$.
> To further clarify, see footnote 4: replacing $R$ with $R^{+}$ in VariBAD policy gradient would potentially reduce variance. The VariBAD paper doesn't discuss this at all, and we wanted to provide a complete treatment of our approach.
>
> 2. Let us first consider the case without function approximation. In standard (non meta) offline RL, the offline RL algorithm can ‘stitch together’ parts of trajectories in the data in a manner that leads to high reward. While these trajectories are different from the trajectories in the data, they visit states that were observed (i.e., no extrapolation). In our case, we are **planning in belief space** - the offline RL algorithm can stitch together trajectories in the hyper space. Therefore, the offline meta RL algorithm will similarly be driven to visit belief states that were observed (and lead to high reward).
> When function approximation is introduced, we need to consider the generalization error (we never really visit the same hyper state twice). We investigate this point thoroughly - please see Figure 7 and the discussion following it in the supplementary material. Note also that the training data is diverse (Figure 11), as it contains trajectories from complete RL training logs. Given enough diversity, we observe that the VAE generalizes well (Figures 7, 10).
> Finally, a well-known issue in offline RL with function approximation is that high Q values can be given to actions that were not visited in the data, leading the planning to choose ‘out of distribution’ trajectories. We investigate this in Table 1, where we evaluate the use of CQL (a SOTA offline RL method) to train the critic network in our method.
>
> 3. This is a positioning issue - please see our general comment.
>
> 4. Please see footnote 7, where we reference HER and clarify the difference. In HER, relabelling addresses an exploration problem with sparse rewards. In our case, reward relabelling (which is different from HER) addresses MDP ambiguity.
>
> 5. It is easy to construct an example where MDP ambiguity enables a completely uninformative belief model to fit the data perfectly (actually, Figure 7 displays exactly such a case). Therefore, without additional assumptions (on the MDP hypothesis set, or on data collection), performance could be arbitrarily bad. This is one of the most important implications of our work.
>
> 6. We compare our results to both (online) PEARL and (offline) MBML. MBML is the *only* offline meta RL algorithm we are aware of (prior to our work), and it is evaluated using the exact data as our method. We cannot think of a fairer comparison.
> Whether online meta-RL is harder or easier is difficult to answer. In most offline (non-meta) deep RL studies, the offline setting proved much harder, and to our knowledge the top performing deep RL algorithms in both continuous and discrete benchmarks are online. In our experiments, we did not observe a clear winner between MBML and PEARL, but our BOReL significantly outperformed both.
> Furthermore, as we state in the Experiments section, in our comparison we aim to show that BOReL can approximately learn a Bayes-Optimal policy. It is trivial that online VariBAD will obtain better results than BOReL since it is not restricted to offline data. We therefore chose MBML and PEARL as baselines since they use a conceptually different exploration strategy (Thompson sampling).
> We do provide results of online VariBAD in Figure 12 (both conventional, and our better performing off-policy variant). Comparing online performance to offline performance is not informative, as it depends mostly on the quality of the offline data.

---

> > ### Comment · Reviewer_YKZ2 · 2021-08-23
> > **Regarding Point #1**
> >
> > Thank you for the detailed clarifications.
> >
> > With regards to 1. could you detail how this differs from the "root-sampling" strategy introduced by Guez et al for the purposes of MCTS planning in POMDPs/BAMDPs [1]. They show that for the purposes of planning under a belief (regarding states or MDPs) it is sufficient to sample from that belief and run simulations according to that sample without posterior updates. In particular, there is a proof ([2] pages 96-98) that such an approach produces the correct (posterior) visitation distributions at each tree node which seems to be closely related to what you are describing?
> >
> > Many thanks!
> >
> > [1] Guez, A., Silver, D., and Dayan, P. (2013) Scalable and Efficient Bayes-Adaptive Reinforcement Learning based on Monte-Carlo Tree Search. JAIR
> > [2] Guez, A. (2015) Sample-based Search Methods for Bayes-Adaptive Planning. Ph.D. Thesis, University College London.

---

> > > ### Author Response · Authors · 2021-08-25
> > > **Response to Reviewer YKZ2 (2)**
> > >
> > > Thank you for your response.
> > >
> > > **As we note explicitly in footnote 3**, Proposition 1 is indeed similar to the “root-sampling” approach by Guez, Silver and Dayan, but applied to our particular context (offline meta-RL instead of online MCTS).
> > > For the sake of the review process, we would like to further clarify our contribution here.
> > > This paper presents the first offline deep BRL algorithm. While the technical content of Proposition 1 is not new (and a reader proficient in BRL would probably not find it surprising), we wanted to give a complete theoretical motivation of our algorithm. In this respect, Proposition 1 formally connects the VariBAD approach and offline RL. We emphasize that this discussion does not appear at all in the original VariBAD paper, and not all readers would make the immediate connection with the work of Guez et al, so we do think it is an important part of the paper.
> > >
> > >
> > > To give yet another example why this delicate point is important, consider the following thought experiment, which is based on the discussion in [1]. Let us forget for a moment the belief state. The policy gradient approach, as used in VariBAD, could be applied to the states and rewards just as well, and in this case would seek the (locally) optimal policy based only on the state observation. While this policy would not be Bayes optimal, the algorithm is sound (see [1] for a proof). The same cannot be said about value-based off-policy algorithms such as Q-learning, which require Markov state transitions. Applying Q-learning to states (and not hyperstates) does not make any sense in the BRL/POMDP setting. All of this is just to emphasize - the explanation in the original VariBAD paper is not complete enough to simply replace its online RL component with an offline one.
> > >
> > >
> > > [1] Safe, multi-agent, reinforcement learning for autonomous driving. Shalev-Shwartz et al., 2016

---

### Official Review · Reviewer_cUbW · 2021-07-16

**Rating:** 5
**Confidence:** 3

**Summary:**

The paper studies the problem of Offline Meta-Reinforcement Learning (OMRL). In this setting, one observes M trajectories coming from N different environments sampled from the same underlying distribution. The goal of the agent is to perform well in expectation on a new unseen MDP drawn from the same distribution. To do so, the authors propose relabelling strategies to mitigate the problem of MDP ambiguity they introduce/formalize and to enable off-policy learning for the VariBad algorithm. The resulting algorithm was tested and compared with PEARL (online) and MBML (offline) on a series of environments with discrete or continuous action spaces, and sparse or dense rewards.

**Limitations And Societal Impact:**

There is no social impact to consider in this work.

**Main Review:**

**Pros**

- The problem investigated in this work is novel and will certainly gain importance in the future. The paper poses the problem well and qualitatively discusses its challenges, lying the groundwork (MDP ambiguity) for future contributions.
- The relabelling strategies seem simple to implement and efficient in mitigating the MDP ambiguity introduced in the paper.
- I liked the different environments used, with both sparse and dense rewards highlighting the difference in performance when using the relabelling strategies.

**Cons/Questions**

My main concern regards the positioning of the paper as doing offline RL. The setup certainly is but the solution isn't.

- The policy relabelling assumes that one can access the learned policies and that additional data can be collected. The offline RL setup would be greatly simplified if one could make such an assumption (in practice). I acknowledge that the authors clearly stated this issue -  L235-237: "We importantly note that changing the data collection method in hindsight is not suitable for the offline setting. Therefore, the proposed scheme should be viewed as *a guideline for effective OMRL data collection*." Nonetheless, the paper is still advertised as doing offline RL.
- The reward relabelling suffers the same limitation. The authors assume that the reward of transitions can be relabelled. This requires access to the reward signal. I agree this is a softer assumption as opposed to having access to the full environment dynamic. The authors mentioned this is not a problem - L254-257: "We remark that the assumption on known reward (during training) is mild, as the reward is the practitioner’s method of specifying the task goal, which is typically known; this assumption is also satisfied in all meta-RL studies to date." For meta-RL it might be true, but this is a hard assumption when working in an offline setting.

**Reason For Rating**

I consider this paper as a good stepping stone on which the research community can build. The framing of the problem, as well as the discussion around the ambiguity in identifying the MDP, is well conducted. The experimentation is fair and adequate to highlight the benefit of their approach. However, I would encourage the authors to re-position the paper as the offline setting is violated on multiple occasions in their solution. Despite these, I consider the work to be a valuable contribution to the research community and I could be swayed if my concerns are answered.

**Time Spent Reviewing:**

7

---

> ### Author Response · Authors · 2021-08-09
> **Response to Reviewer cUbW**
>
> Thank you for your supportive review.
>
> We will reposition our work and will change the title to “Offline Meta Reinforcement Learning - Identifiability Challenges and Effective Data Collection Strategies” – we will tone down our claims and will make it clear that our method is not “the solution to all OMRL”, as described in the general statement above.

---

### Official Review · Reviewer_Gv9V · 2021-07-17

**Rating:** 7
**Confidence:** 4

**Summary:**

The paper is doing quite a few things at once:

1. Proposing an algorithm for learning effective exploration policies for meta-RL from only offline data
2. Defining the offline meta-RL problem (not technically new but concurrent with other recent work)
3. Defining the MDP ambiguity problem, which proves troublesome for inference-based meta-RL algorithms, and providing two baseline approaches to solving it

The authors accomplish 1 by converting the VariBAD to work with off-policy data and 3 with intuitive strategies for additional data collection or augmentation, which provably address the problem. The experiments suggest that the proposed solutions are necessary to solve problems for which identifying the MDP is difficult.

**Limitations And Societal Impact:**

Some discussion of safety during exploration for RL/meta-RL systems would be welcome. In addition, the authors do not describe the technical limitations of their work. In particular, describing settings for which the proposed data collection/augmentation strategies are not practical would be worthwhile.

**Main Review:**

Overall, the paper is generally well written and focuses on an interesting, important problem. The solutions presented are intuitive and well-motivated, and the experimental results match the theoretical claims. My primary concerns are (a) the impracticality of the data collection/augmentation solution to MDP ambiguity and (b) the explanation of the ambiguity problem and how identifiability is connected to overlapping identifying states.

### Writing

The writing is clear. A concise and informative recapitulation of necessary technical background information needed to understand the problem is given; the Background section has pedagogical value.

One confusion I had re: L149: since R^+ and P^+ are unknown (the original data comes from regular MDPs), isn't it somewhat unsurprising that the VariBAD objective doesn't refer to these explicitly?

### Novelty & Significance

The paper describes an off-policy variant of an existing algorithm, VariBAD, as well as two data collection/augmentation approaches to solving the MDP ambiguity problem. While the solutions are not particularly surprising, but they are well-motivated and explained clearly. Overall, the contributions are sufficiently novel and their effectiveness suggests they are significant as well.

### Experimental Evaluations

The experimental evaluations are generally intended to assess whether the proposed algorithm learns useful exploration policies as well as whether or not MDP ambiguity presents a problem in practice. The experiments are reasonably convincing relative to other papers in this area, although several environments are very simple.

### Remaining Questions

Overall, I think this is a solid paper that probably deserves acceptance on the basis of clear motivation, well-motivated solutions, and reasonably convincing experiments. Again, my biggest point of confusion is regarding the connection of identifiability of data (used in the theory) to the (not clearly defined as far as I can tell) notion of "overlapping states." I also have some concerns that the proposed data relabeling/augmentation approaches are not viable in practice. In order to raise my score, the authors could convincingly answer a few questions:

- Can the authors more clearly define what "overlap" (e.g. L214) means and how it is connected to identifiability? Right now, there is a gap in my understanding of how the identifiability is linked to the specific experimental domains presented, because the identifying states are defined in terms of *pairs* of MDPs, not single MDPs.
- Although the authors mention that constraining the hypothesis space of MDPs is not desirable, this approach would be attractive because it does not require additional data collection/augmentation. Can the authors comment on the viability of such an approach to solving MDP ambiguity problem, or conjecture what such an approach might look like?
- Is extrapolation error not a problem when applying off-policy VariBAD to fully offline data, as is a problem in offline RL?

*Update after discussion period*: in light of the authors' responses to my questions as well as the other reviewers, I have decided to raise my score to 7. I think the characterization of the MDP ambiguity problem is a meaningful contribution that will be of interest to the NeurIPS community, even if the proposed solutions to it are relatively straightforward. The authors can improve their paper (as discussed previously in this forum) by further explaining the significance of the problem they describe and the practical limitations of the solutions that they propose.

**Time Spent Reviewing:**

7

---

> ### Author Response · Authors · 2021-08-09
> **Response to Reviewer Gv9V**
>
> We thank the reviewer for his insights, and address his concerns.
>
> $\underline{\text{Regarding L149 comment}}$: This is a delicate point. In principle, VariBAD directly learns $R^+$ in the VAE, and you can replace $R$ with $R^+$ by using VariBAD’s reward decoder. Our Proposition 1 shows that it is not necessary.
>
> $\underline{\text{Overlap and identifiability}}$: We will add a formal definition of "overlap" to our paper. Our definition is:
>
> “Consider the setting in Definition 1. For a pair of MDPs $i$ and $j$, we say that a state-action pair $(s,a)$ **overlaps** if it has positive probability under both $i$ and $j$, i.e., $P_{\mathcal{P}_i}, \pi_{\beta}^{i}}(s, a),P_{\mathcal{P}_j, \pi_{\beta}^{j}}(s, a) > 0$”.
>
> Similarly, we explicitly define “identifying state-action” (which is currently part of Proposition 2):
>
> “For a pair of MDPs $i$ and $j$, we say that $(s,a)$ is an **identifying state-action pair** if $\mathcal{R}i(s,a)\neq\mathcal{R}j(s, a)$ and/or $\mathcal{P}{i}(s'|s, a)\neq\mathcal{P}{j}(s'|s, a)$”.
>
> With these two definitions in mind, Proposition 2 can be simply rephrased as:
>
> “Consider the setting in Definition 1. If for every $i\neq j$ there exists an identifying state-action pair that overlaps, then the data is identifiable”.
>
> It is true that identifying states are defined in terms of pairs of MDPs, but note that we define identifiability in terms of a set of $N$ MDPs (Def. 1). To bridge the gap, our Proposition 2 guarantees that if for every pair of MDPs we have an identifying state which also overlaps, then the entire data is identifiable.
>
> In the experiments, some domains are such that it’s very likely that identifying states overlap in the data. This happens in Half-Cheetah-Vel, since the reward both identifies the task (proportional to forward velocity), and is observed everywhere (it is not sparse). In this domain, one cannot confuse two trajectories from two different tasks as coming from the same task with different policies.
> In Semi-circle, this is not the case. Figure 2 illustrates how a single task (reward at both locations) can explain two trajectories from two different tasks, when identifying states don’t overlap. Figure 7 in the supplementary, and the discussion after it, shows how this problem manifests in practice.
>
> $\underline{\text{Constraining the hypothesis set}}$: Consider the Semi-circle domain and assume we have prior information: for each task the reward is limited to a single circular area. We can modify the VAE decoder to account for this (e.g., outputting the mean of a Gaussian distribution over the state space instead of the general neural network function in our implementation). In this case, it is sufficient to encounter a single reward location in order to identify the task, and we will not suffer the MDP ambiguity described in Figure 2. In this work, we chose to focus on the general setting where such information is not available. An in-depth treatment of how to encode prior knowledge into VAE-based belief estimators would not fit within the scope of this paper, but is definitely an important future direction.
>
> $\underline{\text{Extrapolation error}}$:  As an offline method, BOReL’s performance depends on the quality and diversity of the data. Therefore, we provide a “Data Quality Ablative Study” (see Table 1) to examine our method’s performance for different offline data collection strategies. As can be seen, data diversity is instrumental; however, we also show non-trivial learned exploration policies in the low-quality data regime (see Figure 10). We also compare our results with a variant of our method that uses CQL (a SOTA offline RL method) to train the critic network (Table 1) – although it improves performance, it is not beneficial for overcoming MDP ambiguity, which is the main problem we identify in OMRL. We hypothesize that future research in offline RL will benefit OMRL as well.

---

> > ### Comment · Reviewer_Gv9V · 2021-08-16
> > **Author response follow-up**
> >
> > I appreciate the authors' clarifications. A few residual thoughts/suggestions:
> >
> > *It is easy to construct an example where MDP ambiguity enables a completely uninformative belief model to fit the data perfectly (actually, Figure 7 displays exactly such a case). Therefore, without additional assumptions (on the MDP hypothesis set, or on data collection), performance could be arbitrarily bad. This is one of the most important implications of our work.*
> >
> > *[from the response to reviewer YKZ2]*
> >
> > I agree that this is an important consequence of this work, and does not receive sufficient attention in the current text, in my opinion. The authors should highlight this more clearly in the main text.
> >
> > *In the experiments, some domains are such that it’s very likely that identifying states overlap in the data.*
> >
> > I agree with this intuition. I suppose what's missing is some sort of approximate metric for quantifying the extent to which a particular finite dataset has identifiability issues, to link up the binary characterization of identifiability in the theory with the 'messy' nature of the empirical datasets. I do realize this is largely out of scope for this paper.
> >
> > *An in-depth treatment of how to encode prior knowledge into VAE-based belief estimators would not fit within the scope of this paper, but is definitely an important future direction.*
> >
> > While I agree with this and appreciate the authors' response, perhaps the authors could include in the paper a bit of discussion about the limitations of the proposed solutions (and mention that encoding stronger priors into belief estimators are an alternative/complementary strategy to addressing MDP ambiguity). This might also help with the positioning issue raised by other reviewers (i.e. clarifying that the proposed solutions are just one set of approaches to solving the ambiguity problem, among others).
> >
> > As a meta-comment (not factoring into my acceptance decision; I realize that many languages deal with gender and pronouns differently than English), I would kindly suggest the authors consider using gender-neutral pronouns when addressing reviewers (such as 'their feedback' or 'the reviewer's feedback').

---

> > > ### Author Response · Authors · 2021-08-17
> > > **Response to Reviewer Gv9V (2)**
> > >
> > > Thank you for raising our attention to using gender neutral pronouns. We will take it into account in the future!
> > >
> > > We agree with the reviewer about highlighting /  discussing in more detail some aspects of our work, and we are happy to make these simple modifications to the paper.

---

### Author Response · Authors · 2021-08-09
**General comment to all reviewers**

We thank the reviewers for their feedback.

We are excited that the reviewers found our work “intuitive and well-motivated”, “focuses on an interesting, important problem” that “will certainly gain importance in the future”, “a good stepping stone on which the research community can build”, and “a valuable contribution to the research community”. The discussion of MDP ambiguity is “novel and highlights an important avenue for future research” and “the experimental results match the theoretical claims”.

It appears that the **only** major issue is the positioning of our work - **is it fair to claim that we tackle offline meta RL when we prescribe data collection policies / require the reward function during training**? Let us discuss that.

We start with an example. In the medical fields, collecting data using randomized controlled trials is standard, and is motivated by the theory of causal inference, which identifies the limitations of detecting cause and effect from offline data. Thus - prescribing well motivated data collection strategies definitely has practical value.

In offline (meta) RL, the ‘academic’ setting is that $(s,a,r,s’)$ data tuples are given ‘as is’, and the agent must use them without any further knowledge. Our solutions are indeed not suitable for this case. Our investigation identifies, however, that without additional knowledge (e.g., through the hypothesis set of MDPs), MDP ambiguity can make learning an effective solution impossible. It is important to understand this limitation, which our work formalizes. We mention that several recent works (that cited us, and even used our code) tried to deal with this limitation in different ways, for example, using additional online learning.

A different setting, however, is when the data collection process can be designed. There is practical motivation for this flavor of offline (meta) RL - when data collection is slow/costly (think robotics, autonomous driving, e-commerce, etc.), it is desirable to collect data only once, and **any insight about how to best collect the data is valuable**. Another important benefit of offline (meta) RL is the possibility to train many different models on the same data - for hyperparameter tuning / model selection / large scale training. Our methods apply in this setting.
Although we clarified the limitations of our solutions in the text, we agree that we should better position our work. We suggest the following (and welcome to additional suggestions):
1. Change the title to “Offline Meta Reinforcement Learning - Identifiability Challenges and Effective Data Collection Strategies”
2. Modify the introduction to clarify the practical value of our methods, along the lines above.
3. Suggest relevant future research avenues.

---

### Decision · Program_Chairs · 2021-09-27

**Decision:**

Accept (Poster)

**Comment:**

This paper introduces an interesting problem, where the meta-agent is required to quickly learn unseen tasks given some offline data for training tasks. In addition, the paper also formally introduces the MDP ambiguity problem induced by the new problem setup. Although the proposed solution (off-policy version of VariBAD + reward relabelling) is a little bit straightforward, it is a reasonable solution to the problem considered in this paper, and the empirical results are also solid. During the post-rebuttal discussion period, the majority of the reviewers agreed that the formal description of the MDP ambiguity problem and the Bayesian RL perspective is interesting enough to be presented at NeurIPS. Therefore, I recommend accepting this paper.